

# River ice analyses and roughness calculations using underwater drones and photogrammetric approach

Reeta Vaahtera[1], Juha-Matti Välimäki[1], Tuure Takala[1], Eliisa Lotsari[1]

[1]Water and Environmental Engineering, Department of Built Environment, Aalto University, P.O. Box 15200, Tietotie 1E, FI-00076 Aalto, Finland

*Correspondence to*: Reeta Vaahtera (reeta.vaahtera@aalto.fi)

**Abstract.** In the Northern Hemisphere, freshwater ice forms a significant part of the cryosphere during winters. River ice cover strongly affects the hydrology and flow characteristics of northern rivers, and the effect can last for several months a year. The magnitude of this effect is on the other hand dependent on characteristics of the ice, especially on subsurface ice roughness. However, ice-covered areas have commonly remained unexplored due to challenging conditions and difficult access. This study focuses on developing an improved approach in studying river ice by applying cost-efficient underwater drone platform and camera solutions in studying the ice underside. Furthermore, the developed methodology utilises a photogrammetric approach, Structure from Motion. One key result of the study is a workflow for reconstructing a digital elevation model of the ice underside. It was found that applied photogrammetric approach also enables calculating roughness coefficient for the ice underside. The results of this study show that underwater drones enable studying river ice in more comprehensive and detailed way compared to conventional methods. Additionally, it is noted that applying Structure from Motion in mapping the ice underside can offer feasible approach in determining subsurface ice roughness, which has wider application potential in modelling fluvial processes in subarctic rivers under changing environmental conditions.

## 1 Introduction

Freshwater ice, encompassing lake and river ice, forms a significant part of cryosphere in the Northern Hemisphere during winters (Brooks et al., 2013). Freshwater ice has a large environmental significance as well as wider impacts, such as effects in socioeconomic systems (Brooks et al., 2013). Sui et al. (2010) describe that in cold climates during winter hydrology of rivers is strongly influenced by river ice. Ice cover alters river hydraulics such as flow patterns and increases the complexity of fluvial processes (Smith et al., 2023; Sui et al., 2010). Compared to open channel streamflow, in ice-covered conditions the hydraulic resistance in the flow is increased and thus the conveyance of the stream decreases (Ghareh Aghaji Zare et al., 2016). Ice-covered flow is significantly different compared to open-channel flow (Demers et al., 2011). In open-channel conditions, the highest flow velocities are typically close to water surface whereas in ice-covered channel the point of highest velocity is lowered closer to the channel bed (Lotsari et al., 2017). In other words, as described by Demers et al.





(2011), the vertical velocity pattern in ice-covered conditions has a parabolic shape. This difference on the other hand is

highly depended on the characteristics of the ice cover (Sui et al., 2010). Ice thickness and roughness determine its effect on the flow and the combination of the two is referred to as ice topography (Lotsari et al., 2019). Subsurface ice roughness has following impact on the flow pattern: the rougher the ice the closer to channel bed the maximum flow velocity is lowered (Sui et al., 2010). Ice roughness depends on the type, formation and age of the ice-cover and the flow conditions (Beltaos et al., 2013). In general, the higher flow rate, the smoother the ice underside (Beltaos et al., 2013). Understanding subsurface

ice roughness of rivers is critical in understanding the impacts of river ice on hydraulics and fluvial processes in ice-covered conditions (Ehrman et al., 2021).

Hydraulic roughness can be expressed as Manning's roughness coefficient n and it can be either back-calculated from hydraulic measurements or calculated based on physical properties of the ice (Beltaos, 2001). The first comprehensive approach for estimating hydraulic roughness of ice appears to be Nezhikhovskiy (1964) whereas Beltaos (2001) has

developed this methodology further and describes that thickness profiles of ice jams can be used in quantifying roughness of their undersides. Then again, Li (2012) has proposed similarly that subsurface ice roughness can be expressed as function of roughness height of the ice underside. This approach has been used for computing reference values for Manning's coefficient for model ice in a publication by Jafari and Sui (2021). Literature values for Manning's coefficient for different surfaces are presented in Table 1. Similar values for subsurface ice roughness have been used in previous publications where the range

has been defined as 0.012–0.03 based on literature review and during model calibration (Kämäri et al., 2015; Lotsari et al., 2019). In field surveys, subsurface ice roughness has also been determined qualitatively based on visual observations (Demers et al., 2011; Lotsari et al., 2017 and 2019).

Publications presenting studies of ice-covered flow have been mainly laboratory studies or flume experiments whereas there have been only few field studies (Lotsari et al., 2019). Examples of flume and laboratory studies can be found by for

instance Wang et al. (2008) and Sui et al. (2010). However, field measurements are crucial for better understanding of hydrological processes and they for instance provide input data for modelling approaches (Lotsari et al., 2019; Smith et al., 2023). Yet, challenging field conditions generally restrict the data acquisition (Ehrman et al., 2021). Field measurements are challenging and time-consuming due to difficulties in accessing the underside of the ice cover as well as to the study sites in general, lack of suitable methods and difficulties in collecting spatially comprehensive data (Smith et al., 2023; Spears et al.,

2014). Also, Kasvi et al. (2019) have outlined that measuring shallow river environments is typically complicated as the measurement devices are generally designed for deeper water bodies. Subsequently the challenges cause lack of comprehensive data on river ice and related fluvial processes (Smith et al., 2023). Especially methodology related to studying ice roughness has remained scant. Ehrman et al. (2021) describe that at the time, no reliable means of quantitatively assessing river ice roughness over a large area had been developed. When reviewing the literature for this study, apart from



the discussed approaches by Beltaos (2001) and Li (2012) no studies presenting methodology for directly measuring or determining roughness coefficient for the ice underside based on ice characteristics were found.

Remote sensing methods can offer solutions to abovementioned limitations by for instance improving comprehensiveness (Alfredsen et al., 2018). Automated photogrammetric approaches, such as Structure from Motion (SfM) have been found to offer efficient tools in mapping river ice surface as demonstrated by Alfredsen et al. (2018) and Ehrman

et al. (2021) and sea ice as Cimoli et al. (2019) demonstrate. SfM offers feasible and effective tools for digital elevation model (DEM) reconstruction in fluvial and aquatic applications (Carrivick and Smith, 2018). As input data SfM requires multiple images of the feature with sufficient overlapping and good resolution (Westoby et al., 2012). For detailed technical description of SfM see Cui (2017) and Schönberger and Frahm (2016). SfM offers a low cost and feasible approach for obtaining high-resolution spatial data whereas the processing can be computationally very demanding (Westoby et al., 2012).

Underwater environments are even more challenging due to potentially insufficient lightning and refraction in different interfaces related to the waterproof camera housing (Kwasnitschka et al., 2013). Moreover, ice can be difficult surface to map (Spears et al., 2014). Yet, examples of successfully applying SfM in underwater environments and in mapping ice surface and underside can be found (Alfredsen et al., 2018; Cimoli et al., 2019; Ehrman et al., 2021; Javernick et al., 2014; Lochhead and Hedley, 2022).

In summary, there are two main limitations in field studies in ice-covered environments: difficult access to the studied feature (e.g., ice underside) and low spatial density of results gained with conventional methods. Whereas remote sensing methods such as SfM offer solutions to the latter, the difficulties in accessing under-ice environments could be solved by utilising an unmanned underwater vehicle (UUV) also referred to as an underwater drone. Different unmanned vehicles such as UUVs have been developed and deployed in various applications and environments including ice-covered conditions

(Canelon-Suarez et al., 2020; Lund-Hansen et al., 2018). Lund-Hansen et al. (2018) demonstrate that UUVs can be used to obtain measurements across large spatial scales in an area that would be difficult to access in other means, such as ice-covered subsurface sea environments. UUVs can be divided into two groups that differ in terms of operating: remotely operated vehicles (ROVs) and autonomous underwater vehicles (AUVs) (Xu et al., 2019). These vehicles work as platforms for measurement devices and in published research, UUVs have been used as platforms for a variety of equipment such as

cameras, sonars, and water quality sensors (de Lima et al., 2020; Erena et al., 2019; Spears et al., 2014). Lund-Hansen et al. (2018) have stated that ROVs suit especially well to under-ice applications and have been used in various applications in sea environments. Spears et al. (2014) on the other hand have deployed a ROV in ice-covered lake environment. However, as literature was reviewed for this study, no applications of ROVs in ice-covered shallow water conditions such as subarctic rivers were found. Then again, a ROV can be considered more suitable for studying ice-covered river environments than an

AUV as localisation and navigation of UUVs in underwater and especially ice-covered conditions is typically challenging (Dzikowicz et al., 2023; Spears et al., 2014).



In this study, a workflow that results in a digital elevation model of the ice underside is developed based on previous UUV and SfM approaches by combining them in a novel way for usage in shallow water river environment using cost-efficient platform and cameras. To the authors' knowledge, similar approaches have not been implemented before. The results from the workflow are further used to calculate Manning's coefficient based on the reconstruction yielded from the workflow. The developed workflow allows obtaining more comprehensive information about the ice-water interface than conventional point measurements and observations. Additionally, it allows determining the Manning's coefficient based on ice properties rather than measured flow characteristics which enable validating the reconstruction based on reference values for Manning's coefficient. The determined coefficient could be used furthermore for instance in modelling approaches.

**Table 1. Suggested ranges of Manning's coefficients for subsurface ice roughness and channel bed roughness.**

|  | Type of surface | Manning's n |
|---|---|---|
| | Sheet ice, smooth | 0.008–0.012[a] |
| | Sheet ice, rippled | 0.01–0.03[a] |
| Ice cover | Sheet ice, fragmented | 0.015–0.025[a] |
| | Frazil ice, new | 0.01–0.03[a] |
| | Frazil, ice aged | 0.01–0.02[a] |
| Riverbed | Main channel | 0.025–0.150[b] |
| | Mountain stream | 0.030–0.070[b] |

[a] (USACE Hydrologic Engineering Center, 2024) ; [b] (Chow, 1959)

## 2   Test and study sites

### 2.1   Aalto Ice and Wave tank

Initial testing of the UUV equipment setup was conducted in ice test basin facilities in Aalto Ice and Wave Tank at Aalto University campus. The size of the water basin is 40 x 40 m$^2$ with 2.8 m depth. It includes a system for generation of model ice, and it is designed for testing maritime structures (Aalto University, 2023). Photos of the test basin and model ice are shown in Fig.1. From this study's viewpoint, it offers good facilities to test equipment in less challenging conditions than the field study site, especially as there were no previous approaches to follow. During the initial testing, approximately 7–10 cm thick ice cover was on the water basin. Ice cover was generated through a spraying process, and it had been in use for other tests prior to testing the ROV, and thus there was a cross-sectional cut in the ice cover which can be seen in Fig. 1.



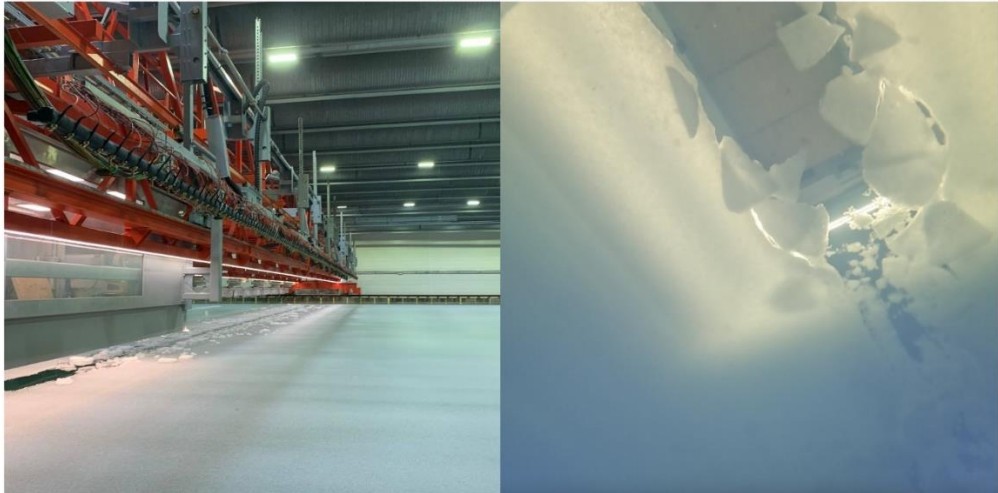

**Figure 1: Aalto Ice and wave tank with generated model ice on 19 January 2023. Picture on the right is taken with the ROV below the model ice cover.**

## 2.2 Pulmankijoki River

The study area, Pulmankijoki River, is located approximately 500 km north from the arctic circle and flows from Utsjoki municipality in northern Finland to Norway. Pulmankijoki River is part of Tana River basin which covers smaller watersheds from Finland and Norway and has a total area of approximately 16 000 km$^2$ (Finnish Environment Institute, 2023). The part of Pulmankijoki river on the Finnish side of the border, referred here as Upper Pulmankijoki, is around 34 km long and flows to north from Lake Ylä-Pulmankijärvi to Lake Pulmankijärvi (Finnish Environment Institute, 2016). The flow continues further into Tana River with the length of 10 km on the Norwegian side (NVE, 2013). The exact study site, which has an area of roughly 16 x 6 m$^2$, locates in a meander bend of Upper Pulmankijoki close to the southern end of the Lake Pulmankijärvi (Fig. 2).

According to classification by the Finnish Meteorological Institute (FMI) northern Finland is classified to the northern boreal climate class (Kersalo and Pirinen, 2009). The study site belongs to the most northern part of this climate class which then again belongs to the starkest vegetation class characterised by short and slow growth season and partly tundra vegetation according to classification by FMI (Kersalo and Pirinen, 2009). The data used in this study was collected in mid-winter conditions in February 2023. The thermic winter 2022–2023 began in mid-October as mean temperature lowered below 0°C and the spring began in early April as the mean temperature rose to 10°C (FMI, 2023a). All in all, thermic winter 2022-2023 lasted around six months. The dates are similar to previous years. Thus, Pulmankijoki is ice-covered every winter and remains partly or fully ice-covered for roughly half of the year (Lotsari et al., 2019). In the weather station closest to the study site, Utsjoki Nuorgam, the average temperatures in February in previous ten years have varied between -4°C and -14°C (FMI, 2023b). Between years 2013–2022 average rainfall in February has been 26 mm, as measured by FMI (2023b).



According to FMI (2023b), the average snow depth in the same years in February varied between 45 and 76 cm. During the data acquisition at the study site, on 22 February 2023, temperature was -29°C and snow depth was on average 29 cm.

The discharges of Pulmankijoki vary significantly between seasons: during spring snowmelt event between late-April and early June (Lotsari et al., 2019), the discharge can be as high as 50 m$^3$s$^{-1}$ and in summer it lowers to around 4 m$^3$s$^{-1}$ whereas in autumn and winter the discharge is lowered even more (Lotsari et al., 2019). According to Lotsari et al. (2022), discharges of Pulmankijoki between years 2016–2021 in February have varied roughly in a range between 0.2–1.4 m$^3$s$^{-1}$. Measured discharge in February 2023 varied in range of 0.849–1.323 m$^3$s$^{-1}$ when measured at different cross-sections at the study site. The discharge of 1.323 m$^3$s$^{-1}$ was measured at that cross-section where the ROV was placed in the water (Fig. 2). Measured ice thickness at the study site was approximately 40 cm. According to Lotsari et al. (2019), Pulmankijoki River represents a typical subarctic river considering its flow characteristics and river ice processes. Generally, the water quality in rivers in northernmost Lapland is classified as excellent (Finnish Environment Institute, 2019). Measured turbidity at the study site in was on average 0.8 FNU which can be considered as almost clear water, and therefore suitable for applying photogrammetric approach.

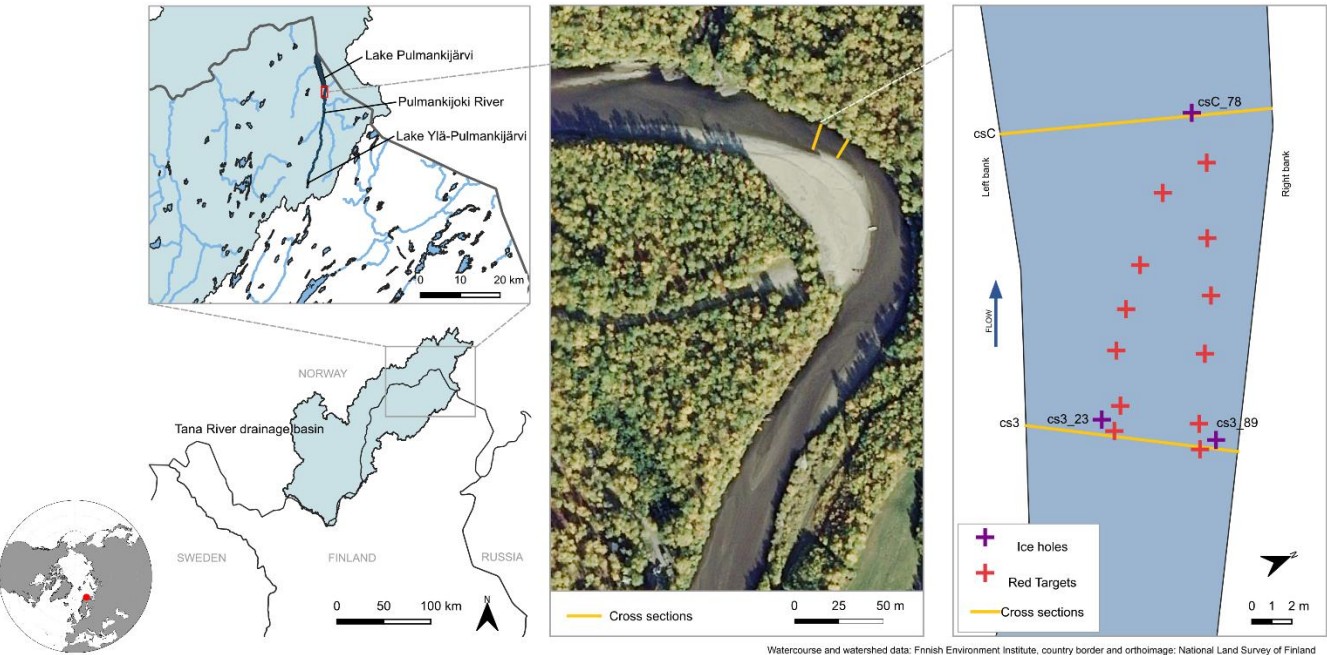

**Figure 2: Pulmankijoki River study site located in northern Finland. Exact study site is located in a meander bend and the studied area covers roughly 16 m long area between two cross-sections (marked with yellow). Ice holes and targets used in the reconstruction of a digital elevation model of the ice underside are marked with purple and red in the rightmost map. The ROV was placed in water ice hole cs3_89 in cross-section cs3.**



## 3 Data and methods

### 3.1 Equipment and initial testing in Aalto Ice and Wave Tank and in the field

A cost-efficient professional ROV platform Chasing M2 Pro was utilised for collecting videos under river ice. The ROV has eight thrusters that allow movement in all directions, and it is operated via a cable (i.e., a tether) with maximum horizontal radius of 400 m (Chasing, 2022). The ROV includes a built-in 4K/1080p camera, and it is possible to mount different equipment such as additional cameras on it. Built-in camera and external cameras were used in field surveys. Two GoPro

cameras (HERO10 Black) were mounted on each side of the ROV for the data acquisition. The ROV platform and the used camera setup is show in Fig. 3.

Before the field surveys an initial testing of the equipment was conducted in laboratory conditions in Aalto Ice and Wave Tank to test the setup and planned settings. The initial testing included setting up the ROV and different targets with known geometry and carrying out a few test drives while taking videos. Targets with known geometry and location and

attached control points were placed to test their usability, e.g., whether they are visible and can the control points be recognized in the ROV videos. Control point approach is similar to what is used in aerial photogrammetry (control points referred to as ground control points [GCPs]) as presented for instance by Over et al. (2021). For these test drives with the ROV, one GoPro was mounted on the ROV as an external camera, and videos were taken with both built-in camera and the external camera in Aalto Ice and Wave Tank. Eventually in field conditions two GoPros were attached on ROV. Collecting

video data from which the input frames can be extracted was assumed to be the best way of fulfilling the requirements of SfM processing, as key requirement for SfM input data is sufficient number of well-exposed and overlapping photographs of the feature of interest as described by Westoby et al. (2012). Visual inspection of the videos revealed that the videos taken with external cameras appeared of better quality (c.f. Fig. 4). GoPro videos appeared sharper or clearer and to have higher contrast compared to the built-in camera. Additionally, the external cameras allowed adjusting the angle in which the videos

are captured whereas the built-in camera captures only forward. Different angles were not tested in the initial testing and hence, different settings were used in the field study as described later in the text.

The initial video settings of the GoPro cameras used in the Aalto ice and Wave Tank were selected based on suggestions from the camera manufacturer and previous research applications and the same settings were applied in the field. The final parameters and their justification for the best quality data acquisition and for analysing under-ice topography (SfM method)

are presented in Table 3. For underwater photography, the setting combination by camera manufacturer is either 4K resolution, 60 FPS (frames per second) and wide lens, or 1080p resolution, 240 FPS and wide lens since the settings of the used GoPro model allow either high resolution or high frame rate (GoPro, 2021). In SfM applications the number of detected points of interest or keypoints is dependent on the image resolution – high resolutions will yield the highest number of keypoints (Westoby et al., 2012). Similarly, Over et al. (2021) also note that images with low resolution may not be fully



supported. Regardless, Westoby et al. (2012) notes that higher resolution might not necessarily be better as it might lead to lengthy processing times. Additionally, GoPro (2021) suggests high frame rates to be used to capture fast action. Hence, the combination of high frame rate (120 FPS) and lower resolution (1080p) was chosen (Table 3). Image stabilisation (HyperSmooth by GoPro) was set off based on the recommendation from SfM software producer Agisoft (2023a). The wide lens setting allows 16–34 mm zoom level of which the non-zoomed 16 mm was used. However, wide lenses cause more lens

distortion, yet the used SfM software automatically aims to correct this distortion (Agisoft, 2023a). Then again, in underwater photogrammetry the effect of lens distortion is smaller due to refraction, which is also automatically calibrated in the SfM software (Shortis, 2019).

Also, Westoby et al. (2012) note that the robustness and battery life of the equipment must be considered especially in remote areas or difficult conditions. Despite the low temperatures and otherwise challenging environment, the camera and

ROV batteries were sufficient for the field measurements although the extreme conditions posed some challenges such as the thrusters getting frozen very easily. Yet, all measurement had to be done efficiently and consecutively without lifting the equipment up from water to ensure sufficient battery life and avoid freezing.

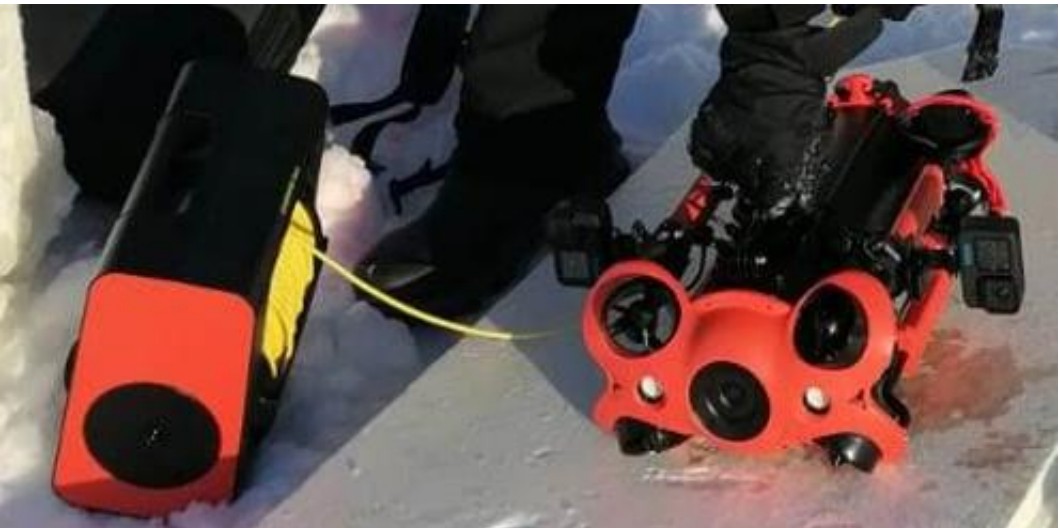

**Figure 3: The ROV platform (Chasing M2 Pro) and the used camera setup.**



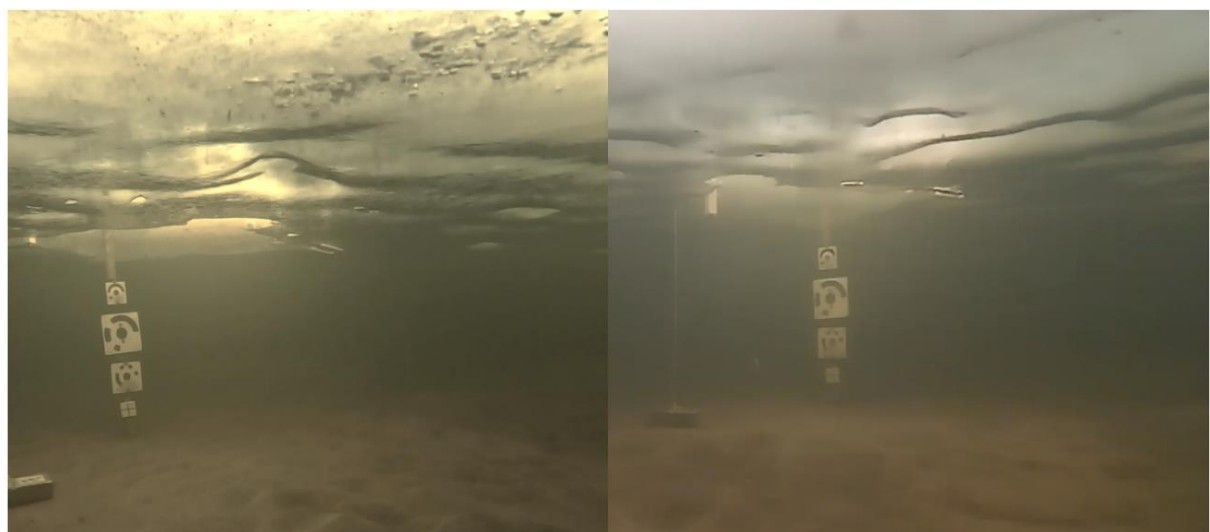

**Figure 4: Comparison of image quality between frames taken with GoPro (left) and built-in camera (right) from approximately same location and time step. These two frames presented here have been edited to make them more comparable: the brightness of GoPro frame has been increased and both have been cropped as the GoPro frames were originally taken with portrait orientation and the built-in camera captured with landscape mode. The frames used in the SfM-processing were not cropped. The fields of view differ due to differences in cameras and their orientation.**

## 3.2  Data acquisition and georeferencing

The ROV platform and mounted GoPro cameras were applied in collecting video material for further processing in an area between two cross-sections (Fig. 2). Both cross-sections consisted of a line of drill holes (⌀ 200–250 mm) which allowed access to the underside of the ice cover. Few drill holes were sawed to bigger, diamond shaped holes in the ice, that were used for placing control points and the ROV underneath the ice cover (see Fig. 5). These are referred to as ice holes, and they are marked with purple in Fig. 2. All together seven cross-sections had been drilled, and based on the characteristics of the channel, the area between cross-sections 3 and C (see Fig. 2: the names of the cross-sections follow the naming of Lotsari et al. [2022]) was selected. The reason for selecting this area was its location in the deepest area among all cross-sections and its varying bed topography: the flow depth was higher at the outer bank and lower at inner bank. It was assumed that the varying topography could potentially influence the ice cover as well.

Localisation and navigation present challenges in underwater environment and ice cover even increases these challenges (Spears et al., 2014). This approach aimed to overcome the issue by manual localisation of the ROV (backtracking the path) and georeferencing the reconstructed elevation model using control points attached to targets with known location. In the study site, two types of targets were placed under the ice: vertical poles reaching from ice surface to riverbed with multiple control points attached, and right-angled targets (combination of wood and angle irons; referred now on as "angle targets") that allow placing the flat control point to the underside of ice (see Fig. 5). These were both placed into the ice holes in cross-sections 3 and C (see Fig. 2), detailed placing is presented in Table 2. One pole was also placed through one of the



smaller drill holes in cs3. Additionally, in total 12 driveway markers (orange plastic poles), referred to as "red targets", were placed through the ice cover between the two cross-sections with known distance from the top of the ice surface, and resulting in approximately 10 cm below the ice cover. Exact locations of the control points attached to targets were calculated from RTK-GNSS (Realtime Kinematic Global Navigation Satellite System) measurements which were taken above the ice cover as the points below the ice cover were impossible to measure directly. In the early data processing steps,

locations of the control points were calculated based on differences in x, y, and z-coordinates as the dimensions of the targets were known. For the vertical pole targets, it was assumed that the location of the control point can be calculated by only changing the z-coordinate according to known length of the pole. Measured coordinates for the angle targets required corrections in all three directions. For these so-called angle targets, the calculated locations of control points can be expressed as

$$\begin{cases} \text{E}_{computed} = \text{E}_{measured} + \Delta\text{E} = \text{E}_{measured} + \cos\left(\frac{\pi}{2} - \psi\right) * d \\ \text{N}_{computed} = \text{N}_{measured} + \Delta\text{N} = \text{N}_{measured} + \sin\left(\frac{\pi}{2} - \psi\right) * d \end{cases}, \tag{1}$$

where E and N are the easting and northing coordinates, $\psi$ is yaw i.e., direction of the target compared to north, and $d$ is the horizontal length of the angle target. Yaw was converted from measured degrees to radians and is subtracted from $\frac{\pi}{2}$ to modify the angle so that the same equation works for all types of angles (acute, obtuse, and reflex).

Eventually, ROV was placed into the water from the rightmost ice hole in cross-section 3 (Fig. 2). It was driven under the ice cover between the two cross-sections through varying paths aiming in capturing as comprehensive images of the area and as many control points as possible. Five ROV drives with different GoPro camera angles were taken. The path of the ROV was kept as close to previous runs as possible but due to the lack of positioning system and currents the paths changed slightly. The mounted GoPro cameras were first set to shoot directly forward, secondly 45 degrees up, and then directly

upwards. All these settings were applied as it was only after the measurements possible to see, which setup of the camera angles was best in the prevailing depth and lightning conditions. Both cameras were set to capture in same angles during each drive. Afterwards, the videos captured with the upwards pointing setting were found to be the most suitable for further processing to capture subsurface ice roughness. When the camera was mounted to point directly forward, both channel bed and ice cover were visible. The camera was pointing 45 degrees up and directly upwards, only ice cover was visible. When

the camera is pointing 45 degrees up, it made the texture and shape, e.g., "bumps", in the ice cover more visible compared to videos taken directly upwards. Based on these observations, the SfM reconstruction was experimented using videos captured with camera pointing 45 degrees up and directly above. However, it was noticed that the image alignment and keypoint detection worked significantly better with the latter. Hence, the videos taken with upwards pointing GoPro camera were used as input data for the elevation model reconstruction. In the take with camera pointing upwards, the ROV was driven from ice

hole cs3_89 to csC_78 (Fig. 2), doing a loop around that ice hole and then going back to the starting point. Since real-time





positioning of the drone was impossible under ice with the available equipment, the path was backtracked from the videos according to the targets.

**Table 2. Detailed placing of different target types.**

| Cross-section | Drill hole | Target type | Number of control points |
|---|---|---|---|
| cs3 | 8–9 | Angle | 1 |
| | | Pole | 2 |
| | 5 | Pole | 2 |
| | 2–3 | Angle | 1 |
| | | Pole | 2 |
| csC | 7–8 | Angle | 1 |
| | | Angle | 1 |
| | | Pole | 4 |




**Figure 5: Target setup used in the data acquisition. a) Photo of a diamond shaped ice hole (hole csC_78, see Fig. 2) where two angle targets are attached. Photo is taken facing towards upstream and also ten driveway markers referred to as red targets are visible in the photo. Note that in the final setup, also vertical pole targets and in total 12 red targets were placed. b) Two types of targets placed under ice (ROV image) and schematic pictures of c) angle target (wood and iron) with horizontal metal plate attached to allow a control point to be placed to the underside of the ice cover and d) vertically positioned pole with four attached control points. Illustrations are not to scale.**



**Table 3. Used settings for GoPro HERO10 Black cameras and their selection criteria for analysing the ice underside.**

|  | Value | Selection criteria |
|---|---|---|
| Frame rate | 120 FPS | High frame rate is suggested for capturing movement and for underwater photography.[a] |
| Resolution | 1080P | Highest resolution does not allow the use of high frame rate. Suggested resolution for underwater photography.[a] |
| Lens | Wide lens | Suggested setting for underwater photography.[a] |
| Stabilization | Off | GoPro image stabilization works by tightly cropping the image.[a] It is suggested not to change the geometry of the original images (e.g. cropping, resizing, rotating).[b] |
| Camera angle* | Directly upwards | Visual check showed that the ice was well visible and photo alignment in Metashape was successful. Similarly, photos taken directly above are recommended for aerial SfM.[c] |

*Final setting selected in processing steps based on field measurements; [a] (GoPro, 2021) ; [b] (Agisoft, 2023b) ; [c] (Over et al., 2021)

## 3.3 Elevation model reconstruction

The obtained video data were further used for elevation model reconstruction using SfM approach with Agisoft Metashape
Professional (version 2.0.1). Later in the text, terms elevation model of the ice underside, reconstructed elevation model and reconstruction refer to the resulting SfM reconstruction. A schematic diagram of the eventual workflow is presented in Fig. 6. The model reconstruction step was preceded by pre-processing the data into suitable input data for the software. The software does not require hard pre-processing of the images and it is even recommended to avoid modifications of the photographs such as cropping or geometrical transformations (Agisoft, 2023a). Hence, the needed pre-processing from raw
data (videos) into suitable input consisted mainly of extracting the video frames into stills in jpg-format.

Although frame rate of 120 FPS was used to capture comprehensive frames of the features, it was noticed that very large numbers of input frames lead to lengthy and demanding SfM processing. The video used for model reconstruction eventually consisted of almost 40 000 frames thus it was not feasible to use each frame. This problem was overcome by eventually selecting only every tenth frame instead of all 120 FPS for SfM model reconstruction. Every tenth was selected as it enabled
lighter processing without having to compromise in comprehensiveness and percentage of overlap of the frames.



Nevertheless, higher frame rate would be potentially useful if paired with more sophisticated pre-selection of frames so that only the ones of good quality (considering visibility, resolution, and disturbances) would be used in further processing. This is also supported by Agisoft (2023a), as they state that in underwater conditions due to additional challenges, it is better to take extra frames rather than having poor data. Frames were extracted from the input video using Python OpenCV library.

Then, the extracted frames were manually inspected briefly to see if the control points are well visible in the images and that the overall quality of extracted frames seems sufficient. However, no actual pre-selection other than extracting only every tenth was conducted as the number of frames was too high to select frames in an efficient way.

The SfM processing was then conducted mostly based on the workflow described by Over et al. (2021). Processing in Metashape started with setting software preferences and importing the extracted frames into the software. Coordinate system

ETRS-TM35FIN was used as the project coordinate system and for marker reference, since it was the coordinate system used in RTK-GNSS measurements. Measurement accuracy for marker coordinates, referred as marker accuracy in Metashape, was set to 0.01 meters based on expected GNSS measurement accuracy. The term marker refers to the assigned locations of control points in Metashape. When importing all frames at once, it was noticed that the software does not work efficiently with approximately over 2000 frames, and hence the model was built in two separate and partly overlapping

chunks. Minimum requirement for georeferencing the model is at least three visible control points, and the recommended number is ten (Over et al., 2021). Hence, to be able to georeference the model there must be at least three control points well visible in the imported frames. Yet, if there were too many frames (roughly over 2000), the software was not able to process them. It was essential to find a balance between achieving sufficient spatial coverage and enabling efficient model reconstruction. Eventually, the final model was built as a combination of two separately reconstructed chunks. After

importing the frames, markers were assigned to the control points in frames and respective calculated RTK-GNSS locations were assigned to them. This can be done either manually or automatically and, in this study, it was conducted manually due to the challenging lightning conditions in the input frames. First, an image of each individual control point was searched, then markers were assigned followed by assigning correct coordinates for each marker. If a control point was visible from e.g., different direction, the same marker was assigned to multiple images to improve their recognition. Brightness of the

frames used for setting the markers was adjusted to be able to see better the exact centre of control point. Eventually control points from two angle targets and one pole were used.

Following step was to align the images which refers to automatized processing where camera locations are computed and tie points are recognized based on the input images (Agisoft, 2023a). The alignment process took multiple hours per chunk. This could be due to challenging surface texture of ice as the object texture can affect quality of the model and for

instance untextured and transparent surfaces can be challenging (Agisoft, 2023b). After alignment, the input data section was checked for any unaligned images. Agisoft (2023a) recommends setting markers manually to the unaligned images to indicate same features than in the aligned images and then retrying the alignment step. However, by experimenting it was



noticed that with the used input data the unaligned frames were impossible to align in the same project, which could potentially be due to lack of keypoints. Hence, unaligned images ended up being discarded. This step was conducted

separately for both chunks. After alignment, camera alignments were optimized with a built-in command which performs a bundle adjustment (BA) on the aligned data. For more detailed description of BA, see Schönberger and Frahm (2016).

Subsequently, a dense point cloud was constructed in Metashape based on the aligned keypoints and depth map computed from keypoints. Dense point cloud is a denser and more comprehensible point could of the mapped feature when compared to keypoint cloud as demonstrated in Fig. 8. This step was again repeated separately for both chunks. After the

dense point cloud was constructed, the point cloud was manually inspected in case there would have been clearly inaccurate and distinguishable points that do not correspond to the real feature. These points, such as separate and non-attached small clusters of points were manually deleted. The processing steps presented this far result in a scaled reconstruction of the feature in local coordinates results. For referencing the whole model into a global coordinate system, at least three of the control points should have at least two projections in the built model meaning that in the alignment that point should be

recognized in at least two images. In this approach, the guided approach for marker projection was used, meaning that the markers do not need to be set manually to each photo where they are seen but the software automatically detects all images where the point is visible. It was first checked that at least three of the previously assigned markers were included in the built model and when these requirements were fulfilled, automated georeferencing was conducted. This was done again separately for the two chunks, which as a result were eventually in the same coordinate system and ready to be combined as one

complete reconstruction. First the separate chunks were aligned using built-in point-based method in Metashape that according to Agisoft (2023a) aligns chunks by matching photos between the chunks. Other option would have been marker-based alignment where the chunks are aligned based on marker locations (Agisoft, 2023a). Since the two chunks had only three markers both, the point-based method was assumed to perform better. After alignment, the chunks were merged which then yielded the final elevation model of the ice underside.



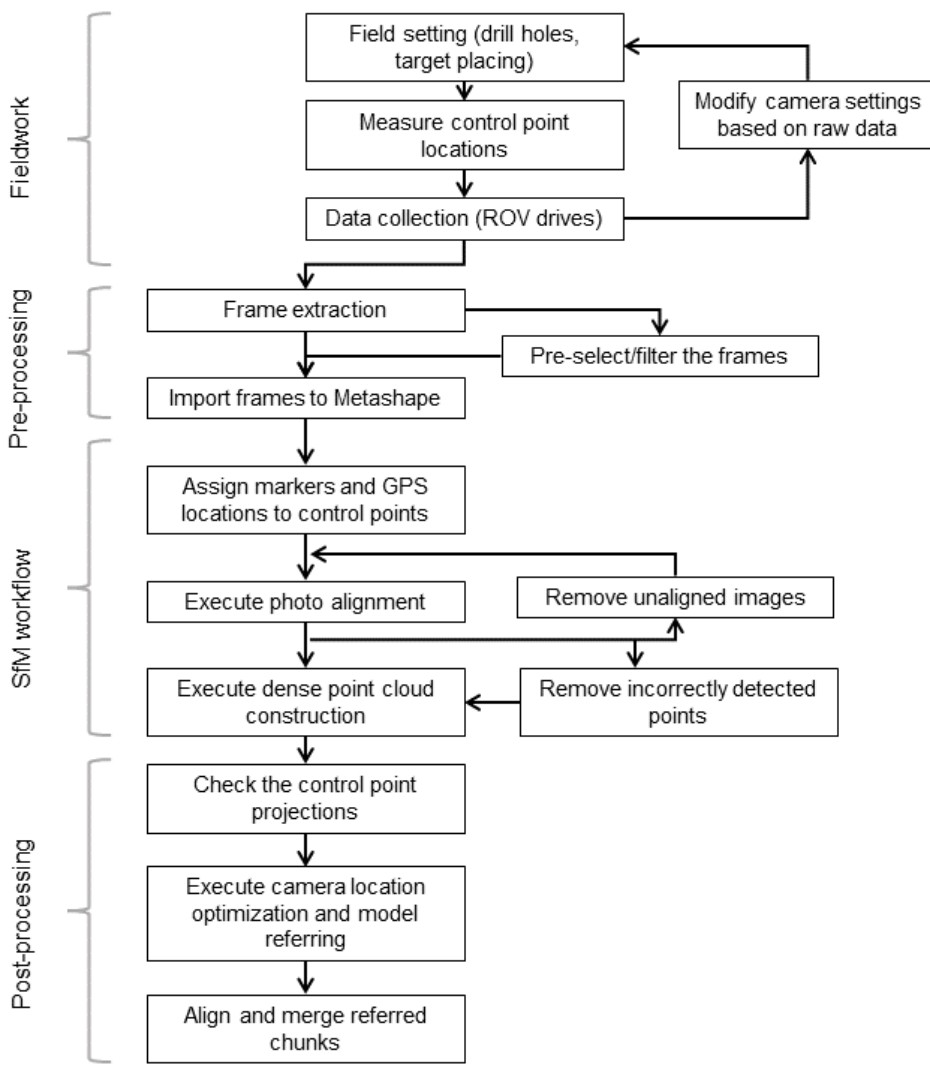

**Figure 6: Schematic diagram of workflow for SfM processing.**

### 3.4 Ice roughness based on the reconstruction of under-ice topography

The subsurface ice roughness was estimated based on the roughness height derived from the reconstructed elevation model. Average roughness height was determined as standard deviation of height (z-coordinate) based the approaches used by Smart et al. (2004) and Momeni et al. (2023). Roughness height can be expressed as standard deviation as follows.

$$k_s = \sqrt{\frac{1}{N}\sum_{i=1}^{N}(x_i - \overline{x})^2} \; , \tag{2}$$



where $N$ is the count of measurements, $x$ is single measurement and $\overline{x}$ is average of all measurements. The roughness heights of the ice underside were determined using surface profiles extracted from the reconstruction. Furthermore, the roughness height is converted into Manning's coefficient using approach by Li (2012) according to which this coefficient can be expressed as

$$n_i = 0.039 * k_s^{\frac{1}{6}}, \tag{3}$$

where $n_i$ is Manning's coefficient for ice and $k_s$ is average roughness height of the ice underside. Li (2012) has presented that roughness height $k_s$ can be determined based on flow measurements but also as physical feature of a surface. The total workflow for determining Manning's coefficient is presented in a schematic diagram in Fig. 7. In the beginning, the georeferenced elevation model of the ice underside was exported from Metashape to CloudCompare (version 2.13.0) where its orientation was manually corrected so that height coordinate is perpendicular to the horizontal plane. This step is dependent on user as it is done manually and hence, it would be beneficial to avoid corrections at this point if the initial orientation is acceptable.

Next the model was divided into cross-sections. Cross-sectionally derived average roughness heights are used in the calculations in this study due to two reasons. Firstly, this approach would have allowed feasible removal of single cross-sections that seemed to be biased or otherwise not representing the subsurface ice roughness. Note that in this study, eventually no removal of cross-sections was found necessary. Secondly, as the whole reconstruction wasn't horizontal but slightly curved, roughness heights derived from the whole reconstruction wouldn't have corresponded to real conditions and the use of cross-sections aimed to minimise the potential errors due to the reconstruction not being a straight plane. Also, to minimise errors, the reconstructed elevation model was segmented into four segments that appeared to be planar and were then modified to horizontal orientation prior to actual cross-sectioning. The horizontally aligned segments were then cut into 239 cross-sections in total. The spacing between cross-sections was set to 5 cm and width of a cross-section was 0.5 cm. Cross-sectioning was done perpendicularly to the longer side of the reconstructed elevation model. These values were selected to get sufficiently dense segmentation for comprehensive results and also similar spacing was used by Momeni et al. (2023). The coordinates of dense point clouds representing each cross-section were exported into text files and surface roughness was calculated based on the variation in the height coordinate in these cross-sections. Finally, average of all cross-sections in one segment was taken to get results that present one horizontal section and can be compared to reference values.



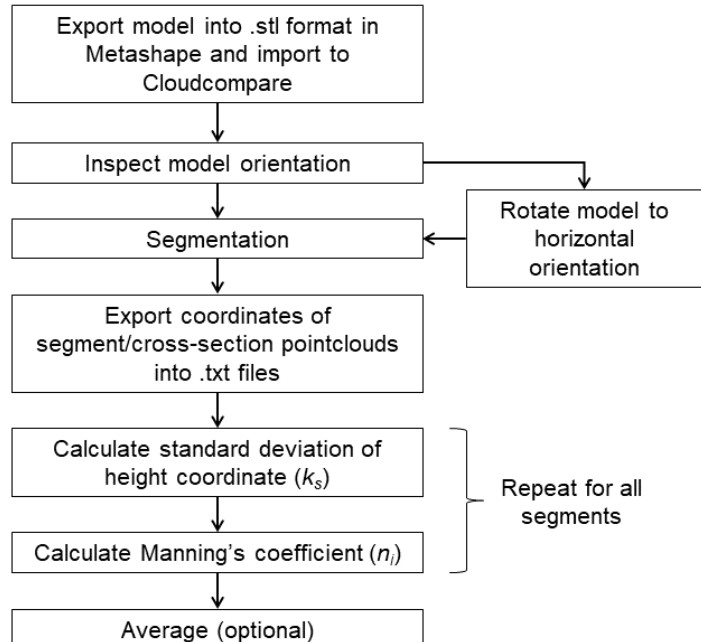

**Figure 7: Schematic diagram of workflow for roughness determination.**

## 3.5 Evaluating success of the methods

380   The obtained results were evaluated visually and by comparing the obtained roughness heights and Manning's coefficients with reference values. Also, the differences between measured control point locations and their estimated locations in the model i.e., control point errors were analysed. The visual inference was simply conducted by comparing observations from the videos with the reconstructed elevation model to see, although very roughly, whether the model geometry corresponds to the real conditions and observed ice topography. For instance, Lochhead and Hedley (2022) have used similar method to

385   assess results from underwater SfM applications. Then again, the roughness coefficient values derived from the reconstructed elevation model were compared to reference values for Manning's coefficient. Comparison between the calculated roughness coefficients and reference values is presented in Table 5 as root mean square error (RMSE) values. RMSE can be expressed as

390   $$\text{RMSE} = \sqrt{\frac{\sum_{i=1}^{N}(predicted_i - actual_i)^2}{N}} \ , \qquad\qquad\qquad (4)$$

where *predicted* refers to calculated value, *actual* refers to reference values and *N* is the count of values.



As the studied conditions were mid-winter, the reference values for subsurface ice roughness were assumed to be the values given for smooth or rippled sheet ice. Suggested Manning's coefficient values for smooth and rippled sheet ice vary in a range between 0.008–0.03 (USACE Hydrologic Engineering Center, 2024). Similar values are also suggested in previous research applications (Ehrman et al., 2021; Kämäri et al., 2015; Lotsari et al., 2019). Subsequently, according to Eq. (4), the acceptable roughness height varies in the range between 0.007 cm and 21 cm. The range seemed relatively large, and it was assumed that the accuracy of the model was not sufficient for detecting the smallest differences. Then again, Ehrman et al. (2021) have determined that roughness height for smooth ice is mainly below 10 cm and for rough ice below 20 cm. Hence, the range of acceptable average roughness height values was set as range from 1 to 10 cm (mid-winter conditions). For calculating RMSE for Manning's coefficient, value $n_i = 0.02$ was selected based on approach from Lotsari et al. (2019) as their study was conducted at the same study site than this study. The subsurface ice roughness in this study was similar to the conditions in Lotsari et al. (2019) also based on visual evaluation: when following approach by Demers et al. (2011), the subsurface ice roughness in this study can be determined as smooth-rough based on the videos whereas Lotsari et al. (2019) determined the ice in their study smooth-rough as well.

## 4    Results

### 4.1   SfM processing results

An elevation model of the ice underside was produced from the ROV video data. A capture of the reconstruction from directly above (i.e., below the ice cover) and its coverage of the whole study site are presented in Fig. 8. The reconstructed elevation model covered an area of 26 m$^2$ with approximate length of 15 m. It was reconstructed out of 3096 images out of input of 3364 images i.e., 268 frames were discarded. This number of frames did not include all extracted frames of the video used, as some had been discarded earlier. The final dense point cloud consisted of 52 244 668 points. The reconstructed elevation model covered a narrow path between the two cross-sections and a small loop around ice hole csC_78 but did not extend completely from one cross-section to another. Although the input frames extended from cross-section to cross-section, it was found that the SfM processing was not able to align all frames leading to the reconstruction covering only part of the input data.

Captures of the reconstructed elevation model, which aim to visualise the surface texture are presented in Fig. 9. Surface texture of the reconstruction could be described as rough or rippled rather than even or smooth. The surface was comprised of small "bumps" that were not evenly spread (Fig. 9). It was not possible to reliably identify areas with different textures or roughness in the reconstructed elevation model since the whole surface seemed to have this similar texture. Then again, the northwesternmost part of the reconstruction was clearly more characterized by distinguishable shapes shown in Fig. 9 (e.g., the ice hole, targets, and drill holes) compared to the narrow section which did not include any of these. As can be seen in



Fig. 9 the reconstructed elevation model was able to represent the ice hole and two additional drill holes recognizably.
Additionally, it was noticed that the reconstruction allowed identifying small details such as the ROV cable and a rope.

Metashape automatically gave absolute errors for the markers, i.e., control points locations. Marker errors present distance between input location and estimated location based on the reconstructed elevation model (Agisoft, 2023a). However, in this application the only available input locations were the locations of the three control points and corresponding marker errors derived from Metashape are presented in Table 4. The total-values express RMSE of the given
values and they are also derived from Metashape. The total error of x-coordinate was 8.44 cm and for y-coordinate it was 12.11 cm. Considering the surface roughness, which was assessed as variation of height coordinate, the most interesting is the error in z-coordinate. The average of absolute errors in z-coordinates for the three control points was 0.077 cm and corresponding RMSE is 0.08 cm. For both angles the total error was 13.7–13.8 cm and for the pole slightly more, roughly 20.9 cm.

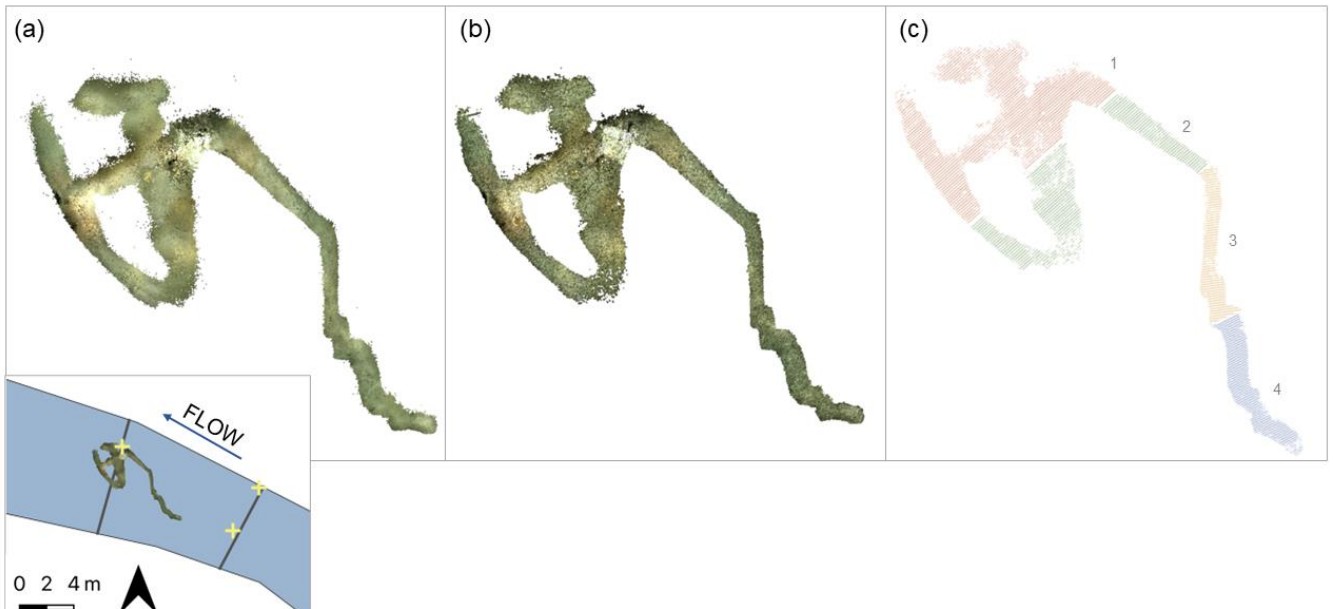

**Figure 8: Capture of the reconstruction taken directly towards the surface (ice underside) and visualization of its coverage of the whole area. From left to right the different versions present keypoint cloud, dense point cloud and cross-sectioned dense point cloud. Different segments in c) are distinguished with different colours and the segment numbers.**



**Table 4. Differences between measured and estimated locations of the control points.**

| Control point/Error [cm] | X | Y | Z | Total (RMSE) |
|---|---|---|---|---|
| Angle 1 | -6.03 | -8.63 | -0.05 | 13.71 |
| Angle 2 | -5.90 | -8.49 | -0.06 | 13.75 |
| Pole | 11.93 | 17.13 | 0.12 | 20.88 |
| **Total (RMSE)** | 8.44 | 12.11 | 0.08 | |

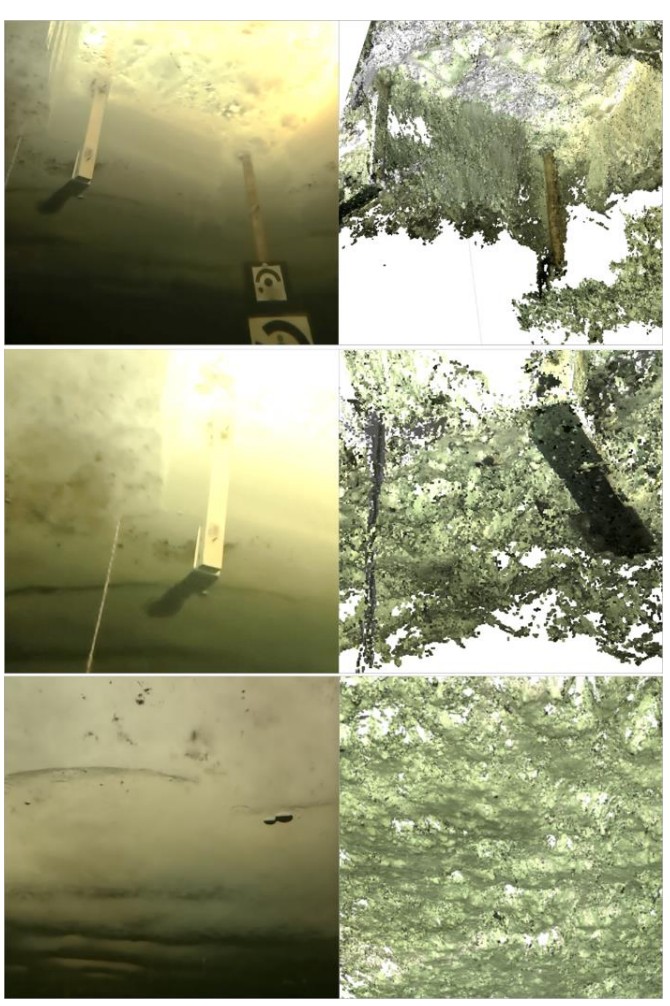

**Figure 9: Comparison between the raw data (left column) and captures of elevation model of the ice underside (right column). As exact ROV locations weren't known, the input frames and captures of the reconstruction are not taken exactly from the same point.**





## 4.2 Manning's coefficient

Calculated values for average Manning's coefficient for the four segments are presented in Table 5. The difference between calculated values and reference values are presented as RMSE (Table 5). The results for roughness coefficient values varied in range 0.0186–0.0232 and corresponding RMSEs varied in range 0.0015–0.0037. Then again, based on visual evaluation,
surface roughness of the reconstructed elevation model varied slightly between different segments. Higher roughness coefficient values were gained when the computations were performed with cross-sections from the northwesternmost part of the reconstruction. These segments encompassed the region that included distinguishable shapes, such as the ice hole, making them less united or cohesive than the other two (segments 3 and 4). However, deviation in the calculated Manning's coefficients was relatively small: around 10 % of the average and thus any cross-sections were not deleted even though the
approach would have enabled this. Based on the reference values for Manning's coefficient and expected accuracy, accepted range of 1 to 10 cm was set for the roughness height. The derived roughness height values varied in a range from 1.2 to 4.5 cm meaning that they corresponded well to the accepted range. Average RMSE value for roughness coefficient was around 11 % of the result and the RMSE values were all ranging from 8 to 16 %, indicating that for all cross-sections, the calculated roughness coefficient is quite close to 0.02. The segmentation and respective cross-sections used for the calculations are
visualised in Fig. 8. Examples of cross-sections and their standard deviations are plotted in Fig. 10 and from the figure, it can be observed that the variation in the height was not uniform and did not follow any clear pattern e.g., evenly spread ripples or bumps. Then again, the mean z-coordinate and standard deviation seem to present well the roughness of cross-sections in question.

**Table 5. Roughness heights derived from the elevation model of the ice underside and respective roughness coefficients and their errors. Segmentation and respective numbers can be seen in Fig. 8. For more detailed description of segmentation see Sect. 3.4.**

| Data | $k_s$ [cm] | $n_i$ (SfM) | RMSE [$n_i$] |
|---|---|---|---|
| Segment 1 (60 cross-sections) | 4.465965 | 0.02323 | 0.003696 |
| Segment 2 (45 cross-sections) | 3.607238 | 0.022418 | 0.002673 |
| Segment 3 (63 cross-sections) | 1.761698 | 0.019894 | 0.001507 |
| Segment 4 (71 cross-sections) | 1.165265 | 0.01857 | 0.001949 |
| **Average** | 2.750042 | 0.021028 | 0.002456 |



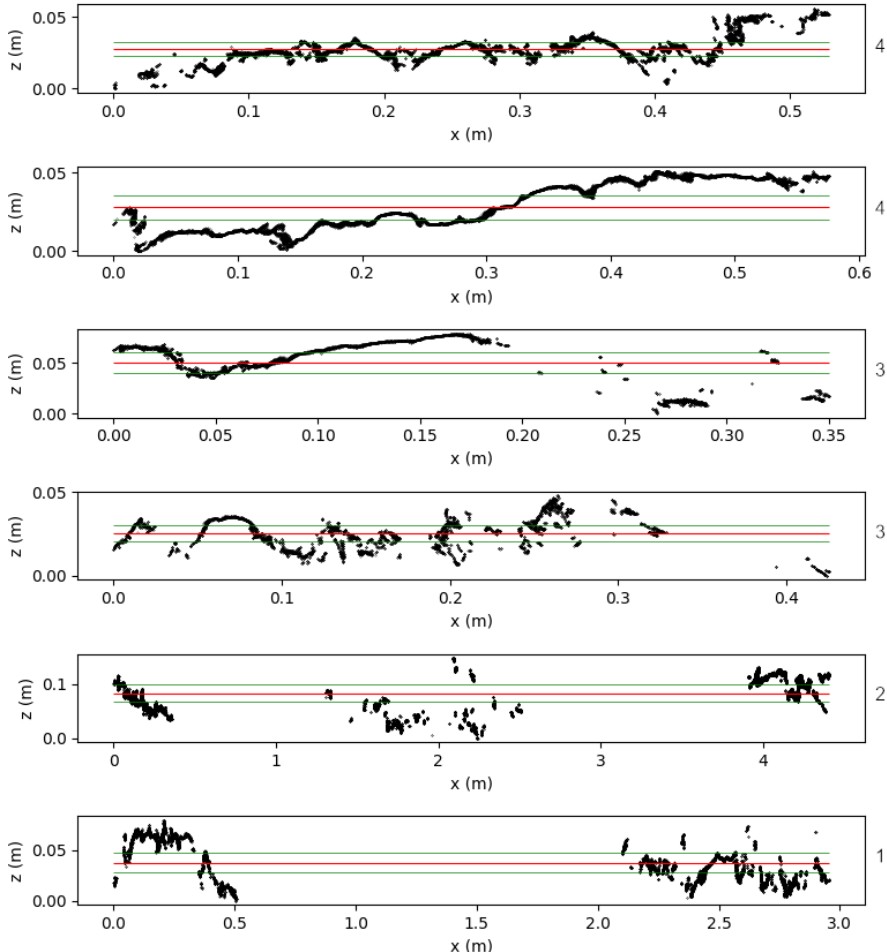

**Figure 10: Plotted vertical coordinates of every 40th cross-section of the final cross-sectioned reconstruction. The topmost graph is from the upstream end of the reconstruction and lowest on the other hand from the downstream end and rest are from between in the same order. The corresponding segments are marked on the right side of each graph. Mean height value (z-coordinate) is marked with red and standard deviation of height with green.**

## 4.3 Overall summary of the developed workflow

An overall workflow is presented in Fig. 11 summarising the developed novel workflow discussed in pervious sections and the required steps and respective outcomes are presented on a general level. The noted limitations and uncertainties associated with each step are listed. Relative weight of each step is illustrated in the workflow. The overall workflow was summarized to eight steps that proceed from finding the settings and practices for data acquisition to finally computing the subsurface ice roughness based on the reconstruction. In between, the different steps of the workflow are as follows: data acquisition, quality check for raw data, selecting the input for SfM processing, SfM reconstruction and georeferencing,



quality check for the elevation model, and cross-sectioning the reconstructed elevation model. The most crucial steps are data acquisition (2nd step) and SfM processing (5th step). As illustrated in Fig. 11, the SfM processing step was the most time-consuming, and especially computationally demanding. Then again, the data acquisition step has the second highest relative weight as the field work is generally time-consuming. Of course, the workload of the two is different compared to each other: processing takes time but is relatively passive whereas the fieldwork can be very intensive. Additionally, the

expected outcomes of these two steps hold especially high importance for the results. Considering the uncertainties presented in the overall workflow, again the data acquisition and SfM processing steps appear important. Both can include uncertainties and limitations that should be considered carefully when following the workflow and interpreting the results.

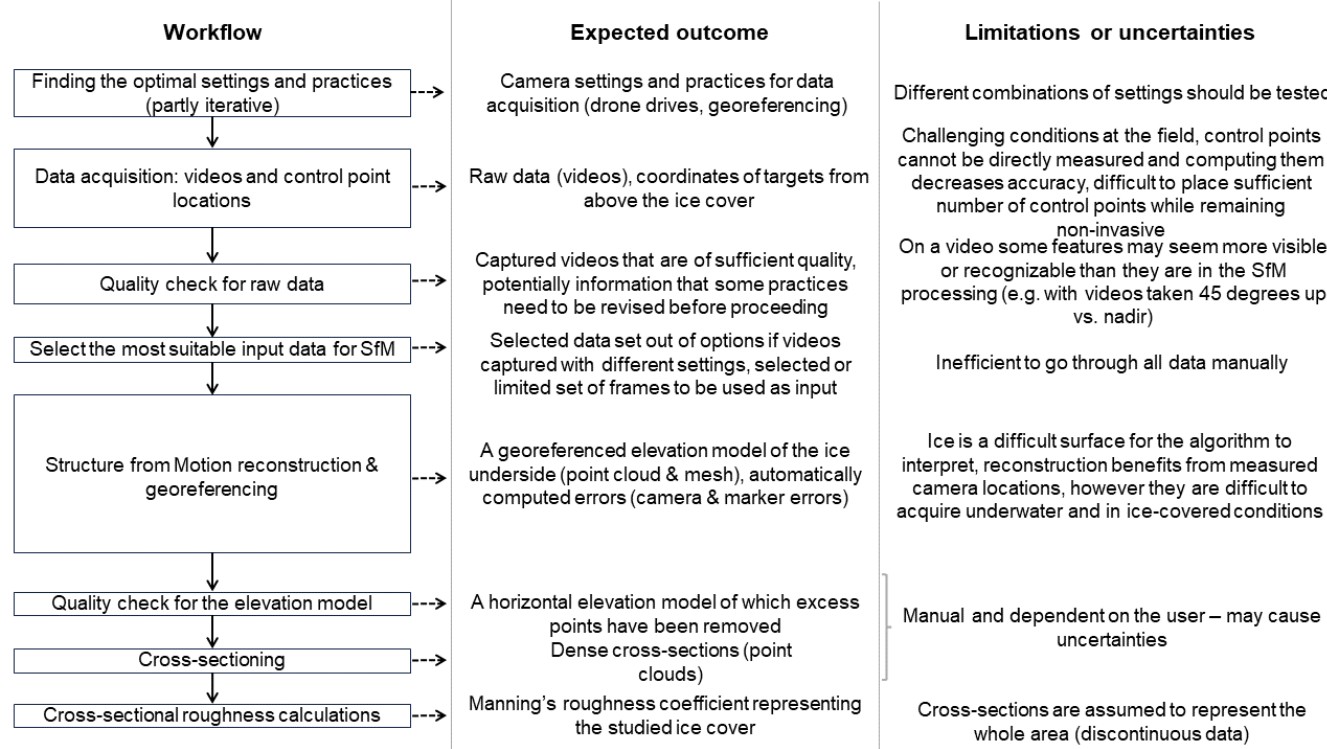

**Figure 11: Overall workflow summarizing the steps required for elevation model reconstruction and calculating Manning's coefficient. Height of the frames outlining workflow steps show the relative weight of each step (considering how time-consuming and computationally demanding they were).**





## 5 Discussion

### 5.1 Optimal practices


Since no previous publications on using ROV platforms and SfM technique in mapping river ice underside were found, the workflow was developed based on related literature and through field trials. The presented optimal practices are mostly related to the first three steps in the overall workflow presented in Fig. 11: finding the optimal settings and practices, data acquisition and quality check for raw data. The data acquisition allowed comparison between different settings considering 500 the camera model, angle of view and the use of targets whereas other parameters are defined only based on previous applications and assumptions.

As discussed, visual inference of the video data showed that compared to videos taken with the built-in camera the GoPro video data are more suitable for the proposed approach due to flexibility in adjusting the camera angles and noted better quality. The differences in video quality and clarity can be caused by differences in waterproof camera housing which 505 is a typical reason of increased distortion and inaccuracies in underwater photogrammetry (Lochhead and Hedley, 2022). Then again, in quality check for raw data and SfM processing steps (see Fig. 11), it was noticed that the SfM algorithm recognizes features best from the videos taken with cameras pointing upwards. It is also suggested in reference material that input photos for SfM should be taken with cameras pointing directly or almost directly (such as near-nadir or high-angle in aerial photogrammetry) to the feature (Kwasnitschka et al., 2013; Over et al., 2021).


The distance between the ROV and the ice cover can also affect the model reconstruction: for instance, Spears et al. (2014) noticed that the slight topography of ice underside was only visible when the UUV was very close to the ice cover. However, no direct suggestions for optimal distance between the camera and feature of interest in underwater SfM applications were found. In general, it could be expected that the closer the camera, the better small features are visible but then again, higher distance between the camera and object would allow better spatial coverage per frame. Obviously, in 515 under-ice applications, also the space between ice cover and channel bed limits the maximum distance. When analysing the quality of raw data and subsequently in SfM processing phase, it was noticed that with the approximate distance used in the experiments (approximately 40 cm) between the camera and ice cover, the features of the ice seem to be distinguishable for the SfM algorithm. Then again, the collected data allowed reconstruction of an elevation model that covers only approximately half a meter wide section of the ice directly above the drone path i.e., the drone should be driven in denser 520 pattern to be able to make reconstructions of larger areas. Kwasnitschka et al. (2013) state that optimal path for a UUV in underwater photogrammetry would be continuous and gridded path which has considerable overlap. The reconstruction doesn't cover the total estimated drone path between the two cross-sections since the frames from the beginning of the path were not possible to align in Metashape. Since higher overlap is expected to benefit the SfM processing (Westoby et al., 2012), gridded and denser path with considerable overlap could also help in decreasing the number of unaligned frames.



To sum up, the distance between ice and camera should be sufficiently small for features and texture of the ice underside surface to be well distinguishable which on the other hand can decrease the spatial coverage per frame. Correspondingly, the drone path needs to be dense enough for sufficient overlap which again leads to longer drives to cover large areas. It was also noted that the drone cable (i.e., tether) can disturb the model reconstruction by causing unnecessary keypoint detection. In the future, it could be advantageous to collect videos with different distances between the camera and ice underside to

iterate optimal balance between sufficient detail and covered area per frame. The optimal distance could be iterated in the quality control for raw data step (step 3 in Fig. 11) by finding the most suitable practice based on acquired data.

The used target setting on the other hand allowed georeferencing the model somewhat well. Especially, the type of target where control point was set to the underside of the ice cover, so-called angle target, was found useful as those were well visible in the videos taken with upwards pointing camera. Eventually, the "red targets" were not used for georeferencing

as they were not visible in the aligned frames, yet they were used in backtracking the drone path manually. In future applications, it would be good to have higher number of angle targets more evenly spread across the whole studied area. For instance, for aerial SfM photogrammetry, minimum of ten of evenly spread control points over the whole studied area is recommended (Agisoft, 2023a). Additionally, it was found that the control points can appear quite dark against the ice surface making it difficult to place the markers. However, adjusting the brightness to five times the initial value made them

visible enough for assigning the markers.

### 5.2 Workflow performance

The developed workflow allowed reconstructing a roughly horizontal plane that appears to present the appearance of the ice underside somewhat reliably. This was noticed when comparing the reconstructed elevation model to raw data i.e., videos from the study site. For instance, there were no remarkably distinguishable shapes in the reconstruction as was expected in

mid-winter conditions. Comparison between the model and single frames from the ROV videos is presented in Fig. 9. The geometry of the model seems to be representative of real features – for instance the shape of the ice holes and targets is recognisable and in scale. The vertical walls of the ice hole appear clearly smoother than the ice underside, which also indicates reliability in the model as the sawed cross-sections of the ice are expected to be smoother than the naturally developed underside. However, otherwise the reconstructed elevation model is quite rippled which might be partly biased

geometry since the ice underside is expected to be generally rather smooth in mid-winter conditions. Also, in the collected videos the ice surface seems slightly "bumpy" (see Fig. 9) which on the other hand is not as clearly recognisable in the reconstructed model.

Also, the performance can be assessed through the numeric errors that were possible to derive. Marker errors (Table 4) can be considered acceptable, especially the error of below 0.1 cm in the z-coordinate indicates that the reconstruction could

detect reliably height of the keypoints. According to Over et al. (2021), the marker errors should not exceed the marker



accuracy which was in this study set to 0.01 meters based on expected GNSS measurement accuracy. The marker error in z-coordinate is clearly below the expected accuracy whereas in x and y-coordinates the errors are close and partly over the limit. Then again, the marker errors, especially for the angle targets, are in similar scale than in the approach by Cimoli et al. (2017) where sea ice underside was mapped using SfM. According to Cimoli et al. (2017) this accuracy is sufficiently high

for retrieving complex topographic features. In addition, similar scale of errors for both angle targets indicated that the geometry of the reconstruction around the ice hole and targets was correct. Yet, the marker errors give information of very limited spatial coverage as there were only three reference points in the reconstruction. However, the error in the z-coordinate was small compared to the range of accepted roughness height values, meaning that the obtained values would align with the expected scale even if the errors were significantly larger in other parts of the reconstruction. Then again, the

calculated Manning's coefficient values presented in Table 5, correspond well to literature values presented in Table 1 and especially well to selected reference value of 0.02 which was selected based on previous applications (see Sect. 3.5.). For instance, Ehrman et al. (2021) have gained similar RMSE results in their approach for determining river ice roughness and considered the errors low. Visual evaluation of the plotted averages and standard deviations (Fig. 10) on the other hand indicate that their use in the calculations is justified as they seemed to represent the general deviation of the height

coordinate well.

## 5.3 Uncertainties and future development

The field conditions were rather optimal for under-ice photogrammetry as the water had very low turbidity and snow depth was relatively low allowing light to penetrate through the snow cover and ice. On the other hand, the extreme temperatures posed challenges such as short battery lives and equipment freezing, and shallow water environment limits the

maximum distance between the cameras and ice underside and increases the risk of the ROV getting stuck. Yet, key uncertainties of the used approach and developed workflow are related to georeferencing the reconstructed elevation model and lack of sufficient reference data. Firstly, as described in previous sections, the locations of the control points were calculated from different measurements which increases uncertainties. Especially for the vertical pole, the assumption of the pole being completely vertical might lead to inaccurate x, y, and z-coordinates if the pole was even slightly tilted.

Corresponding issue can also affect angle targets since their vertical part is also assumed to be parallel to the ice. It is also noted earlier that the number of control points is barely sufficient and for future development, higher number would be recommended. It could also be advantageous to improve the coverage of control points as they only covered small fraction of the reconstructed model. However, as also outlined in the overall workflow, a balance needs to be found between sufficient number of control points and how to remain non-invasive since increasing the number of control points would respectively

mean need for more drill holes.




Then again, the uncertainties in the approach could be, at least partly, overcome if the drone platform included localization system. Measured camera positions, in other words ROV location data, could improve the model reconstruction (step 5 in Fig. 11) as suggested by Over et al. (2021). As described, visual inspection shows that the reconstructed elevation model detects shapes that do not correspond to real conditions and aiding the image alignment with camera positions could help with this. Also, accuracy of the reconstructed elevation model could be assessed with camera errors as well as marker errors if the camera locations were available. Camera errors would represent more comprehensively the accuracy of the model in total compared to the three marker errors that were analysed. Even so, the localization and positioning of UUV platforms in ice-covered conditions has still been identified as challenging (Spears et al., 2014). Moreover, the issue does not only concern ice-covered environments but underwater environments in general as GNNS equipment cannot be used underwater (Dzikowicz et al., 2023). Even though direct use of GNSS is not possible underwater, there are other techniques that can be used for underwater localization such as acoustic positioning and Doppler navigation (de Lima et al., 2020; Robert et al., 2017). Typically, the working principle of underwater positioning techniques relies on either establishing a communication link between the tracked object, such as an UUV and a GNSS receiver located on the water's surface or determining the position of the vehicle at water surface using GNSS and then in underwater environments continuing tracking using magnetic, inertial, and acoustic techniques (Dzikowicz et al., 2023). Commercial underwater localization systems suitable for ROVs are available (Deep Trekker, 2023). Yet, as the approach in this study utilised a cost-efficient platform, uncertainties related to equipment are justified to some extent as additional equipment would have increased the cost. On the other hand, de Lima et al. (2020) have noted that including an accelerometer and a compass on the drone would also enable solving the drone positions in local coordinates. Interestingly, the drone platform employed in this work, Chasing M2 Pro, includes built-in accelerometer and compass (Chasing, 2022). Exploring their potential application in deriving drone location could also be worth looking into.

When it comes to assessing the success and reliability of the results, the absence of a parallel method for modelling the ice underside contributes to the uncertainties. The comparison between literature values and results from roughness calculations is based on assumption that the reference values are correctly chosen and thus representative. The use of parallel method for data acquisition for elevation model reconstruction, such as sonar in addition to cameras would allow assessing the gained results better. For instance, sonar data and SfM reconstruction could be used parallel which would allow comparison between similar data sets as well as using the two data sets for different analysis as demonstrated by Robert et al., 2017. Also, the surface roughness of the ice cover could be determined with a different method to get possibly more accurate reference data. For instance, Momeni et al. (2023) assessed surface roughness using a manual profilometer in addition to evaluating the roughness based on SfM reconstruction. This approach allowed them to assess the reliability of the SfM approach by comparing the results gained with the two methods (Momeni et al., 2023). Of course, in field conditions, it is not possible to access the underside of the ice cover comprehensively to manually measure reference profiles. Yet, in





laboratory conditions, such as the Aalto Ice and Wave tank, it could be possible to first collect video data with a drone and use it for roughness determination and after this, cut out a piece of ice and manually measure profiles as reference. Buffin-

Belanger et al. (2015) have presented an approach where ice underside is accessed at the field by cutting a section of ice and turning it around. They analysed the roughness also via elevation model but did not calculate numeric roughness coefficient from the data (Buffin-Belanger et al., 2015).

## 5.4 Future applications

The developed methods could potentially be used in further applications such as investigating under-ice conditions and

determining parameters for hydraulic modelling. The conducted work shows that the use of underwater drone platforms offers interesting opportunities in studying ice-covered rivers. An UUV platform could be used for visual inspection of areas that are otherwise difficult to access such as an ice-covered river. In general, different unmanned vehicles, e.g., unmanned aerial vehicles, surface water vehicles and remotely operated underwater vehicles, are considered potential emerging platforms in studying water bodies and river basins (Erena et al., 2019). Additionally, the use of UUVs in studying ice-

covered conditions enables potentially lower disturbance to the natural conditions compared to conventional, invasive methods relying on drill holes, as also stated by Lund-Hansen et al. (2018). However, it was noticed in the raw data that the ROV induced additional mixing in the sandy riverbed, if too close to the riverbed. Nevertheless, any disturbances to the ice cover caused by the ROV, such as modifying the surface texture, were not noticed. Furthermore, use of UUVs can allow spatially comprehensive measurements – for instance the model built in this study covers an area of 26 m$^2$ which could not

have been studied as comprehensively from the two ice holes used for placing drone and targets. Nevertheless, achieving large coverage with ROV and SfM approach requires time (Robert et al., 2017). Similar was also noted in this work: improving the coverage of the elevation model of the ice underside would require longer times for both data acquisition and processing.

In addition to UUVs offering feasible platform in collecting video data for visual inspection, the developed workflow

for further processing and analysis offers promising results. These results indicate that the use of UUV platforms together with SfM technique could offer a feasible method for reconstructing a model of the ice underside that can be used in further applications such as determining Manning's coefficient. According to the literature review, Manning's coefficient has not previously been determined directly from the properties of ice underside. Moreover, climate change is emphasising the importance of better knowledge of different hydrological processes, especially in the Artic which is affected more

intensively than rest of the globe (Champagne et al., 2023). Hydrological models offer important tools in evaluating processes in difficult conditions (such as under-ice processes) as well as future scenarios whereas the insufficient availability of calibration data has been identified as key limitation in modelling ice-covered rivers (Champagne et al., 2023; Smith et





al., 2023). Hence, the developed workflow is considered topical and useful guideline for determining more representative parameters in future applications such as modelling effects of the ongoing environmental change.

Although the gained results can be considered good, especially in the light of feasibility and cost-efficiency of the used platform, the proposed methodology requires further development to improve accuracy and representativeness of the elevation model reconstruction. It is also important to consider, whether the developed workflow is applicable to conditions different from this work's study site. The performance of a ROV and photogrammetric approach in underwater environments is dependent on e.g., water quality and temperature and the surface texture of the feature of interest (de Lima et al., 2020;

Spears et al., 2014).

## 6    Conclusions

Although ice-cover has strong impacts on northern rivers during winter, the methodology for studying ice-covered river conditions has had considerable limitations regarding accessing the features of interest and spatial coverage of the obtained results. To fill this gap, this study aimed in developing novel methodology by using cost-efficient ROV and attached

cameras, for studying ice-covered rivers, more precisely subsurface ice roughness. The results show that underwater drone platforms and SfM approach can be applied in analysing river ice and calculating the subsurface ice roughness. A ROV was employed to gather video data, which was subsequently utilized to reconstruct an elevation model of the ice underside using photogrammetric approach, Structure from Motion. This workflow for data collection and processing is one the key result of this study. Furthermore, the workflow was applied to reconstruct an elevation model of the ice underside. This reconstruction

was then used for calculating roughness coefficient values and assessing the success of the developed workflow and other results. Numeric errors were decent, especially considering the challenging subarctic measurement conditions and shallow water river environment, whereas visual inspection showed that the reconstruction is not fully corresponding to expected real-life conditions. The calculated Manning's coefficient values, i.e., subsurface ice roughness, were on average approximately 0.02 with a maximum absolute error of around 0.004.

Then again, the results include uncertainties that stem mostly from the difficult conditions and the need for further development of the practices. According to what is presented in reference material, it was initially expected that analysing the ice underside poses challenges in SfM processing as it com-bines two challenging characteristics: underwater conditions and difficult surface texture. These challenges may have led to the noticed inaccuracies in the elevation model of the ice underside such as detection of non-existent textures in the elevation model reconstruction. The other identified uncertainties

include the inaccuracies in measuring control point coordinates, limited spatial coverage of control points and the lack of ROV localization or positioning. The presented uncertainties could be, at least partially, addressed by modifying the approach in the future by e.g., adding more control points to the field setting and including additional technologies such as positioning system for the ROV and a parallel method for determining the surface roughness. Additionally, the spatial

coverage and quality could be improved by modifying the ROV path. However, this study demonstrates successful use of
cost-efficient platform and cameras.

All in all, underwater drones and related remote sensing approaches can offer feasible possibilities in studying ice-
covered rivers and analysing under-ice conditions. The key result of the study is the developed workflow that allows
reconstructing an elevation model of the ice underside and determining subsurface ice roughness based on the reconstructed
model.


*Data availability*. ROV and RTK-GNSS data used in this study are available upon request from the authors.

*Author contribution*. RV prepared the manuscript with contributions from all co-authors. EL, JMV and TT designed and
collected the field measurements. RV did the data processing and analyses with contributions from JMV. All authors
contributed to the interpretation of results and reviewed the manuscript. EL acquired funding, conceptualised the study and
provided supervision.

*Competing interests*. The authors declare that they have no conflict of interest.

*Acknowledgements*. We would like to thank the Research Council of Finland for financially supporting this work
(DefrostingRivers: 338480; HYDRO-RDI-Network: 337394; Digital Waters [DIWA] Flagship; 359248). In addition, the
work was funded by The European Union – NextGenerationEU Recovery instrument (RRF) through Research Council of
Finland projects Hydro RI Platform (346167) and Green-Digi-Basin (347703). The river-ice related measurements were
initiated at Pulmankijoki River in 2014 under the post-doctoral research project of Dr Lotsari, funded by the Research
Council of Finland (ExRIVER: grant number 267345), and this paper is a continuum in the series of these winter season
studies.

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
