# Peer review of "River ice analyses and roughness calculations using underwater drones and photogrammetric approach"

_EGUsphere, 2024_

## Author Comment (AC2)

The authors describe an approach to map the underside surface of inland water ice as a Digital Elevation Model (DEM) using Structure from Motion(SfM). The images are captured with a Remotly Operated Underwater Vehicle (ROV). The novelty lies in the combination of using ROVs for SfM for mapping the ise underside in arctic inland waters, while SfM as such has already been used by other others for the same application. The authors tested different viewing geometries (camera angles) to determine the optimal configuration (image quality, geometric details). The authors further derive the Manning coefficient from the captured data and, thus, present a method to directly measure ice-induced roughness rather than back calculating it from hydrodynamic-numerical modelling.

It is an interesting approach. The English reads well and the paper is well structured.

The main points of criticism are listed below.

Response: Thank you for your thorough work in providing feedback on our manuscript. Your insights are highly appreciated. In this document, we present responses to your comments and feedback.

The manuscript is overly long. Please try to streamline the paper and restrict yourself to the relevant information. Parts of the methods section, for example, readl like an SfM best-practice tutorial, which should not be the focus of a scientific article.

Response: We understand your point of the manuscript being lengthy and will modify the language accordingly to make the text more concise. We will particularly focus on revising the methods section based on your general comments as well as specific comments provided directly on the manuscript. We will correct the spelling mistakes and consider getting the manuscript proofread again, as referee 1 also suggested revising the text.

The main criticism is the lack of photogrammetric expertize in the entire work. I highly recommend to team up with a specialist in underwater photogrammetry.

Response: We will make the requested corrections to the manuscript and revise it to better align with the expected level of expertise. Our university has a team of experts in the field of photogrammetry, and we also collaborate internationally with colleagues from various universities, including universities in Germany and Norway, among others. We would like to emphasise that not all experts we collaborate with are co-authors of this manuscript. Additionally, please note that we are pioneering in photogrammetry in shallow ice-covered rivers.

While the description of capturing and modelling the ice underside is cleare an concise, it dod not get clear, how ice thickness was determined. I understand that the focus lies on the maping the underside, specifically the roughness of the underside's surface, but I assume that ice thickness plays a role as well. Please comment on that and adapt the manuscript accordingly.

Response: We agree that a description of ice thickness measurements was not included as it was not considered relevant to the presented methods. We will add following sentence to section 3.2: "Ice thickness was measured from the drill holes manually using a levelling rod."

I disagree with using low resolution images and high frame rates as the best parameter setting. I assume that the ROV's velocity was slow (< 1m/s)). Thus, high frame rates will result in overly high overlap and reducing the overlap to a reasonable degree but using higher resolution imagery would be better to my strong believe. I even recommend to use still frame images instead of video (if this is an option for the GoPro 10). Please comment and adapt the manuscript accordingly. At least the vendor's recommendation for using high frame rates for capturing fast action is not applicable in this slow-capturing-motion setting. The ROV's velocity should be reported in the paper.

Response: We understand your criticism on this matter, and we will try to justify our choices in more detail. We will revise the manuscript to describe better the parameter selection in the light of reference material, our experiences as well as your valuable insights and suggestions.

We agree that the high frame rate results in excess overlap, especially as we ended up using only one tenth of the video frames as input for the SfM processing. However, we would like to clarify our selection of higher frame rate and lower resolution. Based on our own and other users' experiences with the exact camera model (GoPro Hero 10 Black), it has been observed that the selection of lower resolution results in better performance in low light conditions. To our understanding this is due to the camera conducting so-called pixel binning which has been stated to improve low light sensitivity. We have noticed that this process aids object detection in the final frames. Moreover, please note that this level of detail is not provided by the camera manufacturer.

Then again, as we selected the lower resolution due to abovementioned reasons, the selection of frame rate was not limited anymore. As you describe and as the description of the processing in our manuscript showed, the highest frame rate results in excess frames. Yet, we have found that the higher the frame rate the sharper the individual frames. To our understanding, this is due to automated selection of shutter speed in GoPro cameras: the automatic shutter speed is on average higher with higher frame rate. As lower shutter speed would increase motion blur, we have noticed and concluded that the individual frames taken with higher frame rate are of better quality even though the total number of them was overly high.

Hopefully with these additional explanations we can provide better explanation of why, to our understanding, a lower frame rate coupled with higher resolutions would not have provided us data with the same or better quality.

What comes to using still frames instead of video, based on our experiments with our cameras, the quality of the frames extracted from videos is better than what would have been gained using still photos. With GoPro Hero 10 the minimum interval for time lapse mode is 0.5 seconds which is too low when driving a ROV in ice-covered shallow water and low light conditions. Using the time lapse mode would not result in sufficient overlap for good data for the SfM processing.

We do not know the exact speed of the ROV, but it is true that it is most probably <1 ms$^{-1}$. As we know the approximate distance travelled based on the reconstruction and the time based on the video length, and considering the varying flow velocities, we get an approximate speed of 0.1-0.5 ms$^{-1}$. To clarify the reasoning behind the selection of settings, we will exclude following sentence from line 186 "Additionally, GoPro (2021) suggests high frame rates to be used to capture fast action.". We will clarify the selection criteria of the settings in Sect. 3.1 and in Table 3 as part of rewriting the whole methods section (c.f. comment 2 on page 1). We will add following sentence to line 161 in the manuscript "According to Chasing (2022) the maximum speed of the ROV is 1.5 ms$^{-1}$ and during data acquisition its approximate speed was 0.1-0.5 ms$^{-1}$."

Georeferencing: The underwater targets are installed through drill holes and are measured with RTK GNSS. While the location above the ice is accurate to the RTK accuracy, it is not stated if the inclination of the poles (where the GCP targets are mounted) was considered. This clearly has an effect on the accuracy of the underwater target accuracy. Please comment. Please also find ideas for the GCP network in the attached commented PDF (i.e. levelled/vertical poles anchored on the ground with multiple targets on it).

Response: We set the poles vertically at the study site, levelled them, and secured them in place. Similarly, the so-called angle targets were set so that the part perpendicular to the ice was vertical, and then they were secured in place with screws. This is why we do not consider the inclination of the poles in our calculations. The thick ice enabled us to level the poles and secure the control point installation so that the inclination of the targets does not affect the accuracy of the coordinates.

Regarding the control point setup, please see our response to your specific comment on page 14 of this document. Also, please note that the primary aim of our study was to demonstrate what is feasible under these conditions. Our study is the first to try this type of approach in the conditions we had at the study site. We installed in total 20 different targets in the study site for data acquisition. Our assumption, based on previous fieldwork and tests in laboratory conditions, was that a higher number of targets (including the red targets) would be visible and usable in the videos and, eventually, in the reconstruction. However, the processing steps revealed that not all frames were aligned, and we eventually had to settle for a barely sufficient number of control points. For instance, Over et al. (2021) state (as referenced on lines 300-301 of the manuscript) that the minimum requirement for georeferencing a model is at least three visible control points, with a recommended number of ten.

We understand that a higher number and density of control points would have improved the quality of the alignment. Thus, our aim is to develop the control point setup further as part of the ongoing development of this approach, as also discussed on lines 580-585 of the manuscript. Your insights on this matter are appreciated and will be considered in future efforts.

GCP accuracy: A setting of 1cm is overly optimistic. I assume the accuracy rather to be at 5cm level (RTK accuracy, potential errors due to inclination of poles/angles). Please consider and comment.

Response: We recognize that we may not have provided sufficient information about how we selected such a good value for marker accuracy, but we hope our explanation here clarifies it better. As described above, the poles were levelled, and thus we rely on the accuracy and precision of our equipment. We took three measurements per control point to achieve as reliable results as possible. The setting of 1 cm was selected based on the precision of our RTK-GNSS measurements. The horizontal precision of the three measurements used in the georeferencing varied in the range of 1.1-1.2 cm, and the corresponding vertical precision varied in the range of 1.6-1.7 cm. On average the precision of the measurements was 1.4 cm. Thank you for pointing this out and we will revise the manuscript so that the marker accuracy is discussed in more detail.

Dense Matching strategy (line 322-339): The entire paragraph reads as a best-practice Metashape tutorial. In the paragraph georeferencing, dense point cloud generation, and strategies for handling blocks which broke into >1 chunks. In general, the clear photogrammetric workflow of relative and absolute image orientation, followed by dense matching, followed by DSM generation is not adherered to. Instead, in this paragraph georeferencing follows dense matching, although GCPs have already been measured previously (thus, the model should already be georeferenced). I suggest that the authors contact photogrammetry experts to make the entire image-to-model pipeline more sound.

Response: We will revise the text to better match the expected style of text in this context. We will streamline the description of the SfM processing and make it more concise. To our understanding, we have followed the general photogrammetric workflow referred to in the comment. However, we agree that the section 3.3 could be improved for better clarity. It is true that the dense point clouds are already georeferenced and hence it should not be conducted again. Similarly, it is unnecessary to align the chunks prior to merging as they are indeed georeferenced. As we revise the methods section in general, we will pay attention especially to this paragraph to ensure that the workflow is presented in clear and refined way.

Roughness calculation: You describe that your block (i.e. point cloud) suffered from a spherical/cylindrical deformation. To calculate roughness, you segmented the data and horizontally aligned the piecewise deformed segments. This entire process could have been automated and optimized by calculating the roughness in a sliding-window approach based on the de-trended height coordinates within the local neighbourhood. If you are only interested in the roughness, and not so much in the absolute shape itself, then this would have speared you the segmentation and "horizontal alignment".

Response: Thank you for your suggestion to automate and optimize the roughness calculation using a sliding-window approach. We agree that higher-level automation would be beneficial, especially if the block covered much larger areas and the deformations were more significant. However, with our data, it was feasible to manually segment the point cloud and correct the horizontal alignment, and hence we did not invest in automating the process further. We will revise the manuscript to include this suggestion as an important point for future research.

Ground Sampling Distance(GSD): This important value (i.e. the size of an image pixel on the ground/ice surface) of every photogrammetric block is not reported. The (local) height accuracy mainly depends on the GSD (rule of thumb: sigma_z=2 x GSD) and, thus, knowing the GSD would enable to draw conclusions on the detectablility of the smallest height devi6ations. In other words, the GSD limits the detectablilty of small-scale roughness.

Response: We will report the GSD in the manuscript by revising the sentence on lines 159-160 as follows: "Two GoPro cameras (Hero10 Black) with an average ground sample distance (GSD) of 0.5 mm were mounted on each side of the ROV." We will also revise section 3.5 based on this comment and the specific comments provided (lines 402-405) to address GSD.

Methods section: The section is overly long and often contains discussion related material (justifications) to a too high degree. Please separate methods and discussion more clearly and streamline the workflow description (short and concise instead of describing every detour).

Response: We understand the criticism on this matter and will therefore revise the entire section.

Results: Accuracy vs precision: You reported the RMS at the three GCPs. Due to the lack of redundancy, the reported values cannot be seen as representative for the absolute block accuracy. In addition, the more interesting measure would be precision, i.e. local errors. These could be derived by comparing the (detrended) relative coordinate differences for the individual markes mounted on the poles (Fig 4). Thus, would yield an estimate of the aibility to derive small-scale roughness.

Response: We agree that the redundancy is low, and it would have been practical to use more control points in georeferencing. However, due to prevailing limitations, we had to use the minimum number of three as defined by Over et al. (2021). We did not measure the locations of any control points directly, as they were underwater and beneath the ice cover. However, we agree that the precision of the GNSS measurements should be included and will add the average precision of the GNSS measurements in the manuscript by revising the sentence on lines 296-297 as follows "Marker accuracy in Metashape, which refers to measurement accuracy for the control points, was set to 0.01 meters based on GNSS measurement accuracy and precision. The average precision of the RTK-GNSS measurements used in the final reconstruction was 1.4 cm." Additionally, as we took three GNSS measurements from each target, we can derive their standard deviations (on average 0.003 m) and will include them in the manuscript as well. The reconstruction of ours does not cover more than the three control points for which the coordinate differences have been reported in Table 4.

Figure 9: It is hard to understand the correspondence (left colum - image; right column - DEM). You do have a georeferenced block, thus, it is possible to exactly locate the images. Please improve the Figure in this respect. The ROV positions are well known via the exterior orientation of the images.

Response: Thank you for pointing this out and we agree that Fig. 9 is confusing. We will update the figure so that the images match the elevation model. We will update the caption as follows "Figure 9: Comparison between the video frames (left column) and captures of elevation model of the ice underside (right column)."

I see it as one of the weak points of the paper that there is no independent reference data. If roughness estimation and shape reconstruction are the primary goals, then it would have been possible to saw out a block of ice after capturing it with the ROV and to measure the ice underside from above (full frame cameras, terrestrial laser scanning, total station). That way, the ability to capture the ice underside with a ROV/camera system could be verified and quantified.

Response: The manual method you described would be possible in some conditions if the aim were to derive the roughness of one location (point measurement). This would also require the ice thickness to be smaller than what is found in mid-winter conditions in a subarctic river. If a block of ice with a thickness of 40 cm were to be removed from the river, it would need to be sawed into small pieces to lift it in the first place. Therefore, in these conditions, it would be difficult, if not impossible, to lift an intact block of ice with sufficient size to reliably determine the roughness. To reliably capture different scales of the roughness pattern on the ice underside, we would need to lift such a large block of ice that it is not feasible. However, as discussed on lines 617-622 of the manuscript, it would be interesting to test measuring the ice upside down in laboratory conditions.

Then again, we agree that our manuscript is lacking independent reference data and hence we will include photo of local roughness profile next to a pole with scale in ice hole cs3_89. The photo can be used to get a point measurement of local roughness height of 2.5 cm. However, note that this way we can only get a point measurement.

**Specific comments**

L22 Spelling

Response: We will add the missing comma and "the" on line 22.

L30 Spelling: highly depended

Response: We will edit the sentence on lines 29-30 as follows "This difference on the other hand is highly dependent on the characteristics of the ice cover (Sui et al., 2010)."

L31-32 Spelling

Response: We will edit the sentence on lines 31-33 as follows "Subsurface ice roughness has the following impact on the flow pattern: the rougher the ice, the closer the maximum flow velocity is to the channel bed (Sui et al., 2010)."

L49-L50 Spelling

Response: We will edit the sentence on lines 49-50 as follows "Examples of flume and laboratory studies can, for instance, be found in Wang et al. (2008) and Sui et al. (2010)."

L51 Spelling

Response: We will edit the sentence on lines 50-52 as follows "However, field measurements are crucial for better understanding of hydrological processes as they for example provide input data for modelling approaches (Lotsari et al., 2019; Smith et al., 2023)."

L60 Spelling

Response: We will add the missing comma on line 60.

L62 Spelling

Response: We will edit the sentence on lines 62-63 as follows "Remote sensing methods can offer solutions to abovementioned limitations by improving comprehensiveness (Alfredsen et al., 2018)."

L63-65 Spelling

Response: We will edit the sentence on lines 63-65 as follows "Automated photogrammetric approaches, such as Structure from Motion (SfM), have been found to offer efficient tools in mapping river ice surface and sea ice (Alfredsen et al., 2018; Ehrman et al., 2021; Cimoli et al., 2019)."

L71 Spelling

Response: We will modify the sentence on lines 70-71 as follows "Underwater environments are even more challenging due to potentially insufficient lightning and refraction in different interfaces related to the required waterproof camera housing (Kwasnitschka et al., 2013)." We will modify the sentence on lines 71-72 as follows "Moreover, ice represents a difficult surface to map (Spears et al., 2014)."

L109 What do you mean by that?

Response: We understand that the statement is confusing. We mean that the importance of testing the equipment in less challenging conditions was even emphasised as we couldn't directly follow any previous examples of applying photogrammetry to mapping underside of river ice. However, we understand that the addition is confusing and unnecessary and hence we will edit the sentence on lines 108-109 as follows "From this study's viewpoint, it offers good facilities to test equipment in less challenging conditions than the field study site."

Figure 1 (caption) Spelling

Response: We will revise the caption as follows "Aalto Ice and wave tank with generated model ice on 19 January 2023. Picture on the right is taken with the ROV from below the model ice cover."

L120 Use a non-breaking space zu avoid separation of number and unit. Please adapt the entire manuscript accordingly.

Response: We will edit the manuscript accordingly.

L124 Spelling

Response: We will add the missing comma on line 124.

L129 Spelling

Response: We will add the missing space on line 129.

L135 Spelling

Response: We will add the missing space on line 135.

L140 Spelling

Response: We will add the missing "the" on line 140.

Figure 2 The geographic location (lat/lon) is missing. Please add it to the figure, the caption. and the manuscript text .

Response: We will add the location of the study site to Fig. 2, its caption and in the text. We will modify the sentence on lines 116-117 as follows "The study area, Pulmankijoki River (69°56'01.2"N 28°02'35.3"E), is located approximately 500 km north from the arctic circle and flows from Utsjoki municipality in northern Finland to Norway."

L162 Spelling

Response: We will add the missing comma on line 162.

L165 Spelling

Response: We will edit the sentence on lines 164-166 as follows "Targets with known geometry and location and attached control points were placed to test their usability, e.g., whether they are visible and whether the control points can be recognized in the ROV videos."

L166 Spelling

Response: We will add the missing "the" on line 166.

L169 Spelling

Response: We will add the missing "the" on line 169.

L170-171 Spelling

Response: We will modify the sentence on lines 169-172 as follows "Collecting video data, from which the input frames can be extracted, was assumed to be the best way of fulfilling the requirements of SfM processing, as a key requirement for SfM input data is sufficient number of well-exposed and overlapping photographs of the feature of interest as described by Westoby et al. (2012)."

L183 True, so why are you using video in the first place and not still images? Or is this only a feature of later GoPro Hero versions? It is definitely available for the GoPro Hero 12. Please comment.

Response: Please refer to response on page 2 in this document.

L186 What is certainly key is the balance between image resolution and overlap. A video at 60fps will provide a useless overlap of far more than 90%. Thus the optimum would probably be a decent overlap of around 80+% and high image resolution

Response: We noticed too that the high frame rate results in overly high number of frames and high overlap. As discussed in the manuscript on lines 283-284, we ended up discarding nine out of ten frames for the processing. Hence, you are right that a lot lower frame rate would have led to sufficient overlap. Although, selecting lower frame rate and higher resolution would not have led to better results based on our experiments with the exact camera model as discussed earlier in this document (see page 2).

L186 But you are certainly driving slow, aren't you? Please report the velocity.

Response: Please refer to page 2 of this document for detailed description.

L187 Due to to the arguments above, I doubt that this is the sweat spot for the parameter settings. Please comment.

Response: Please refer to page 2 of this document for additional justification for the selected parameters.

L226-227 Spelling

Response: We will revise the sentence on lines 225-227 as follows "Additionally, a total of 12 driveway markers (orange plastic poles), referred to as "red targets", were placed through the ice cover between the two cross-sections. These poles were positioned at a known distance from the top of the ice surface so that they extended approximately 10 cm below the ice cover."

L229 Did you also consider the inclination of the poles as this would have a direct effect on the coordinates of the targets under water.

Response: The poles were set vertically and secured in place and hence the inclination does not affect the coordinates. We will modify the sentence on lines 231-232 as follows "The locations of the control points attached to the vertical poles were calculated by only adjusting the z-coordinate according to the known length of the pole, as the poles were set vertically and secured in place." We will also add following sentences starting from line 239 "The angle targets were positioned with their vertical part perpendicular to the ice and secured in place with screws. Consequently, the coordinates of the control point could be determined as described above.

L232 cf comment above.

Response: Please refer to response above.

L237 Again, what about potential tilts?

Response: Please refer to response above.

L249 Spelling

Response: We will revise the sentence on line 249 as follows "When the camera was pointing 45 degrees and directly upward, only ice cover was visible."

L252 Spelling

Response: We will revise the sentence on lines 251-252 as follows "Based on these observations, the SfM reconstruction was experimented using videos captured with camera pointing 45 degrees and directly upward."

L276-277 cf above, non-breaking space

Response: We will revise the manuscript accordingly.

L284 Exactly for this reason, I suggest to invest in higher resolution rather than frame-rate. Cf. comment above.

Response: Please refer to page 2 in this document for more detailed discussion on why higher resolution was not preferred.

L287 I don't see this as a good argument. Generally, less but better images (still frames instead of video!!) would be the better solution.

Response: We understand that the argument is not completely waterproof, and we have provided additional reasoning for our settings in this document and will revise the manuscript. The still frame mode of the camera was not expected to provide sufficiently high frame rate and correspondingly good overlap. Then again, we did not want to use the highest resolution available as based on our experiments it would have decreased the cameras' performance in low-light conditions.

We will remove the following sentences from lines 286-289 "Nevertheless, higher frame rate would be potentially useful if paired with more sophisticated pre-selection of frames so that only the ones of good quality (considering visibility, resolution, and disturbances) would be used in further processing. This is also supported by Agisoft (2023a), as they state that in underwater conditions due to additional challenges, it is better to take extra frames rather than having poor data."

L289 We are not speaking of poor data. But more frames does not mean better data!

Response: Please refer to previous response.

L297 Although I'm not an expert concerning RTK accuracy in the arctic region, I doubt that 1cm accuracy can be reached, especially for the height component. I expect the accuracy around 3cm concerning the measured antenna position. However, for the GCP targets also potential errors dou to the inclination of the poles / angles need to be taken in to account. Thus I assume the accuracy rather to be in the 5cm level.

Response: Please refer to pages 3 and 5 in this document to more detailed description of the accuracy/precision of the RTK measurements. The potential errors due to the inclination of the targets are also discussed in previous responses.

L311 Why only three? This is the absolute minimum for establishing georeferencing and does not provide (reasonable) redundancy.

Response: We understand that three control points is the absolute minimum. However, we had to settle to it in this application since only these three control points were visible in the final reconstruction as not all frames were aligned.

L312 Spelling

Response: We will add the missing "the" on line 312.

L325 Had there been any "inaccurate clusters of points"?

Response: We will revise the sentences on lines 324-327 as follows "After constructing the dense point cloud, it was manually inspected for clearly inaccurate and distinguishable points that do not correspond to the real feature and these points were deleted."

L327 Apparently there were outlier points available. Please streamline the manuscript.

Response: Please see the previous response.

L327 On the last page you reported that you were measuring the GCPs as one of the first steps in the Metashape processing. Thus, after BBA you should not only have scaled but also a georeferenced model. Please be clearer in your description an revise the manuscript accordingly.

Response: We agree that the section 3.3 can be improved. We will revise the entire methods section and pay extra attention to the clarity and conciseness.

L339 The entire paragraph rather reads like Metashape best-practice tutorial than a scientific article. As a photogrammetrist, this is an overly long work-around description. The real problem, however, is the right choice of data acquisition. Cf comments above concerning overlap and video vs single-frame images. I postulate that with the right acquisition strategy, the entire block can be aligned in a single block (chunk).

Response: We will revise the section 3.3 to align better with the expected quality and style of writing. We understand that aligning the whole set of frames in one chunk would align better with the general photogrammetric workflow and it is also something we tried. However, as you mentioned, the limitations of our input data might provide reasoning behind the need for this less optimal approach. Yet, we argue that our selection of settings is justified considering the cost-efficient equipment we wanted to utilise. The selection of settings in data acquisition are discussed in detail on page 2 of this document.

Figure 6 As indicated above, image orientation and georeferencing (i.e. both intrinsic and extrinsic orientation should be perfored before dense matching). I know that it is common practice to shift and scale around the model after image orientation, but especially in your case where you do have GCPs this is suboptimal.

Response: Please refer to response on page 4 in this document.

L359-360 cf. comment above. That's exactly what I mean. If the GCPs are properly considered within the image orientation, the resulting model is oriented with the z coordinate pointing upwards. That's best photogrammetric practice.

Response: We agree that the use of control points in our approach could be improved. The eventual number of control points was not what had been planned, but rather what we had to settle with although we expected higher number of the used targets to be visible in the aligned frames. We have discussed this source of uncertainty on lines 580-585 of the manuscript but we will revise the text so that the need for developing the use of control points is clearly emphasised.

L362 That's the interesting what question: What is necessary so that the orientation is correct after bundle block adjustment? That's a question of fabrication and mounting of GCP, their distribution, etc.

Response: It is true that higher number of control points used in alignment and georeferencing would most likely improve the performance of the approach. Please refer to previous responses regarding the challenges considering the control points. We apologise if we did not understand the question correctly.

L367 If that means that the entire block was deformed in spherical or cylindrical way, then it is an indication that the distribution and/or number of GCPs is suboptimal. Such block deformations are well known in GCP-based bundle block adjustment, thus the question of where to place control points need more attention.

Response: Please refer to previous responses regarding the control point setting.

L371 Again, you did not speficy the GSD of the images. Please do so.

Response: We will report the GSD in the manuscript by revising the sentence on lines 159-160 as follows: "Two GoPro cameras (Hero10 Black) with an average ground sample distance (GSD) of 0.5 mm were mounted on each side of the ROV."

L375 This entire process could have been automated and optimized by calculating the roughness in a sliding window approach and based on the detrended height coordinates. If you are only interested in the roughness, and not so much in the absolute shape itself, then this would have speared you the segmentation and "horizontal alignment".

Response: Please refer to page 4 of this document.

L398 Again, this depends mainly on the GSD. As a rule of thumb, the height accuracy of Dense Image Matching is around 2 x GSD. Thus, if you report the GSD, then the quantitative statements concerning the detectability of small scale variations would be possible.

Response: In our study, the GSD was on average 0.5 mm and hence, the previously presented range of roughness heights can be considered to be within the expected accuracy. We will report the GSD in the manuscript and include it in assessing how small-scale variations can be detected.

L402-405 This (an the sentence above) are rather discussion than methods. material. Please shorten and streamline the methods section and only provide the important information to be able to understand your workflow. Please only provide references to related work and justifications where needed.

Response: We will revise the entire methods section to better align with the expectations regarding conciseness, style, and level of detail.

L412 AfIs this really important?

Response: We agree that the highlighted sentence is not important to the reader and there is a need to revise the first paragraph of Sect. 4.1. We will revise the paragraph as follows

"An elevation model of the ice underside was produced from the ROV video data. A capture of the reconstruction from directly above (i.e., below the ice cover) and its coverage of the whole study site are presented in Fig. 8. The reconstructed elevation model covered an area of 26 m2 with approximate length of 15 m. It covered a narrow path between the two cross-sections and a small loop around ice hole csC_78 but did not extend completely from one cross-section to another. Although the input frames extended from cross-section to cross-section, it was found that the SfM processing was not able to align all frames leading to the reconstruction covering only part of the input data."

L413 IMportantß

Response: If we understand correctly, this comment is related to the comment on line 412. Please refer to the previous response.

L418 Aim only? Doesn't it show the surface texture?

Response: We understand that the choice of words is misleading here, and we will revise the sentence on line 418 as follows "Fig. 9 presents images of the reconstructed elevation model, visualising the surface texture."

L423 Of what? Distinguishable shapes?

Response: Yes, we are referring to the mentioned distinguishable shapes (such as the ice holes). We will revise the sentence on lines 421-423 as follows "Yet, the northwesternmost part of the reconstruction was characterized by distinguishable shapes shown in Fig. 9 (e.g., the ice hole, targets, and drill holes) whereas the texture of the narrow section was more consistent".

L434 There is hardly any redundancy in these values as three points is the minimum for establishing georeferencing of the block (global datum shift, orientation, scale)

For the roughness estimation however, the local errors (i.e. precision) is more important. Some of your control boars have multiple targets on it (Fig 4) . It would be interesting to evaluate the relative precision of the different targets on the same plate. This would guide us   to an estimate how well it is possible to derived small scale roughness.

What about the 12 red targets (Fig 2)?

Response: We acknowledge the limited and barely sufficient number of GCPs. However, this was one of the constraints we had to accept in this approach. Please note that, unlike in open water, we cannot rely on divers to manually place and measure the control points underwater in ice-covered conditions in a shallow river. Hence, we do not have measured coordinates or corresponding precisions of individual control points placed under the ice cover. However, we do have the precision of the GNSS measurements and the standard deviation of three measurements per target, which will be included in the manuscript (see page 5). None of the red targets were visible in the frames that were eventually aligned. Additionally, please note that this work is planned to be developed further by utilizing new equipment, such as an improved drone platform.

Figure 8 Add (a), (b), (c) also in the caption.

Response: We will modify the caption accordingly.

Figure 9 It is hard to understand the correspondence (left colum - image; right column - DEM). You do have a georeferenced block, thus, it is possible to exactly locate the images. Please improve the Figure in this respect. The ROV positions are well known via the exterior orientation of the images.

Response: Please refer to previous response to this comment on page 5 in this document.

Figure 9 i.e. images

Response: Please refer to previous response to this comment on page 5 in this document.

L454 English style: Please re-phrase.

Response: We will rephrase the sentence on lines 453-455 as follows "However, deviation in the calculated Manning's coefficients was relatively small: around 10 % of the average and thus no cross-sections were deleted."

Figure 10 Fig 10 clearly shows that the (local) height precision is much better than the reported 8cm for the GCPs. Please see comments above.

Response: We agree that Fig. 10 shows that for individual cross-sections of the reconstruction, the deviation of height coordinate is much below 8 cm. Please note that the vertical marker error for the three control points was reported as 0.08 cm by the processing software (table 4).

L497 Importatn related work in underwater phiotogrammetry literature is missing (Menna, Maas, Noccerino,etc.)

Response: Thank you for pointing out related work that has not been cited in this manuscript. We will revise the references in our work to include as relevant referencing as possible.

L520 It could also be considered to operate the ROV in different distances to the ice sheet (coarse-to-fine approach)

Response: We agree that different distances between the ROV and ice underside should be tested if possible. However, in our study site the water depth below the ice cover varied from 25 to 107 cm (as measured in csC and cs3). Height of the used ROV platform is around 20 cm and hence, for most parts of the river, it would not be possible increase the distance between the ROV and ice much higher than 40 cm. Please note that the area of data acquisition was in the deepest area at the study site as mentioned in the manuscript on line 215. Also, the ROV should not be driven too close to the riverbed, as the ROV was found to induce additional mixing in the sandy bed, if driven too close (lines 631-632 in the manuscript).

L523 Images from a larger distance could probably help for stabilizing the block.

Response: Please refer to the previous response.

L529 sic!

Response: Please refer to the previous responses.

L530 Multiple distances could probably be optimal.

Response: We agree that different distances should be tested if possible. However, we would like to point out that, under the conditions at our study site, the efficiency of data acquisition plays a major role. For instance, the battery life of the equipment poses additional challenges in temperatures of minus 29 degrees and below. We will revise the sentence on line 530 as follows "The optimal distances could be iterated in the quality control for raw data step (step 3 in Fig. 11) by finding the most suitable distance or combination of distances based on acquired data."

L539 1) The size of the markers needs to be adapted to the GSD.

2) My recommendation is to use smaller targets than the ones shown in Fig 4. One idea would be to use poles with (coded) marker targets mounted (glued) over the entire length of the pole and point in all directions. The pole should long enough that anchoring on the ground is possible. And, importantly, the pole should be leveled, which can easily be achieved from above the ice. The vertical pole will stabilize the horizontal alignment of the entire block. This minimizes drilling effort and might spare the angle targets.

In combination with two cameras (nadir and oblique), this should allow to align and (!) georeference the image block.

Response: We thank you for the insights and suggestions regarding the target setup. We will consider these in the future development of the approach. We did not face any issues considering the size of the control points. It is true that the size of the control points needs to be adapted to the GSD and we tried using control points of different sizes and followed the suggestions from the software.

The so-called vertical poles we used, reached from above the ice cover to the riverbed and they were levelled and secured to place. Angle targets were screwed to the ice after levelling. Based on our experiments in this study and additional tests with similar setup, we have found that the angle targets work better. As mentioned on lines 251-254 in the manuscript, we experimented the image alignment with videos taken with camera pointing 45 and 90 degrees upward but with our dataset, the image alignment performed poorly with the lower angles. Then again, when shooting "nadir" (directly upward) the angle targets were visible in clearly higher number of frames compared to the

control points in the vertical poles. However, in future applications, placing the control points higher on the poles should be considered to improve their visibility even more in the videos.

L544 I'm nuot sure that I understand this correctly. One sees clear shapes in Fig. 10. Please comment.

Response: We agree that the wording can be improved in this context. We will revise the sentence on lines 544-545 as follows "The texture of the reconstruction was similar to what was expected in the prevailing conditions, i.e. smooth-rough ice when following the classification by Demers et al. (2011)."

L551 I see it as one of the weak points of the paper that there is no independent reference data. If roughness estimation and shape reconstruction is the primary goal, then it would have been possible to saw out a block of ice after capturing it with the ROV and to measure the ice underside from above (full frame cameras, terrestrial laser scanning, total station). That way, the ability to capture the ice underside with a ROV/camera system could be verified and quantified.

Response: Please refer to responses on page 5 in this document.

L555 Cf comment above. The redundancy is too low and, thus, the values are not reliable enough.

Response: We understand that the redundancy is low. We will revise the entire paragraph so that although the marker errors are discussed, their low spatial coverage is emphasised.

L556 Cf. comment above. RTK positioning accuracy is rather in the domain of 3-5 cm.

Response: Please refer to previous response on page 3.

L558 Were they reported?

Response: The errors for the three control points used in the processing are presented in Table 4. We will revise the caption as follows "Table 4. Differences between measured and estimated locations of the control points (marker errors)."

L564 cf comment above for testing local errors (e.g. via the multiple markers on the pole)

Response: We agree that the suggested way of testing local errors would offer interesting insights whereas the conditions in our study site would not enable this. Please refer to page 5.

L577 Cf above. Reference for the shape reconstruction can be achieved by sawing out a block of ice and measuring it "from above".

Response: Based on our experiences from this study site, this manual measurement would be very difficult if not impossible. Please refer to page 5 of this document.

L581 I would rather call it unsufficient.

Response: We understand that the choice of words could be better. We mean that three control points is enough for georeferencing to be possible in the first place, not that it would be the optimal number. For instance, Over et al. (2021) state (as referenced on lines 300-301 of the manuscript) that the minimum requirement for georeferencing a model is at least three visible control points, with a recommended number of ten. We will revise the sentence on lines 580-582 as follows "It is also noted earlier that three control points fulfils the minimum requirement but is not optimal and hence higher number would be recommended in future research."

L585 cf comment above.

Response: It is true that adding more control points would not automatically mean need for increased number of drill holes. We will revise the sentence on lines 583-585 as follows "However, as also outlined in the overall workflow, a balance needs to be found between sufficient number and density of control points and how to remain non-invasive."

L596 and visual odometry, SLAM, IMU...

Response: We will revise the sentence to include more comprehensive selection of positioning techniques such as the ones mentioned in the comment.

L607 As stated above, this is one of the main weaknesses of the paper. Sawing out a block of ice and measuring it from above would have much increased the creditability of the work.

Response: Please refer to previous response on page 5.

L619 That's not only possible in the lab but also on site.

Response: Please refer to previous response on page 5.